# RE-IMAGINING MULTIMODAL INSTRUCTION TUNING: A REPRESENTATION VIEW

**Yiyang Liu**[1,2][*]    **James Chenhao Liang**[3][*]    **Ruixiang Tang**[4]    **Yugyung Lee**[1]
**Majid Rabbani**[2]    **Sohail Dianat**[2]    **Raghuveer Rao**[5]    **Lifu Huang**[6]

**Dongfang Liu**[2]    **Qifan Wang**[7]    **Cheng Han**[1][†]

[1]University of Missouri - Kansas City    [2]Rochester Institute of Technology
[3]U.S. Naval Research Laboratory    [4]Rutgers University
[5]U.S. DEVCOM Army Research Laboratory    [6]University of California, Davis    [7]Meta AI

## ABSTRACT

Multimodal instruction tuning has proven to be an effective strategy for achieving zero-shot generalization by fine-tuning pre-trained Large Multimodal Models (LMMs) with instruction-following data. However, as the scale of LMMs continues to grow, fully fine-tuning these models has become highly parameter-intensive. Although Parameter-Efficient Fine-Tuning (PEFT) methods have been introduced to reduce the number of tunable parameters, a significant performance gap remains compared to full fine-tuning. Furthermore, existing PEFT approaches are often highly parameterized, making them difficult to interpret and control. In light of this, we introduce Multimodal Representation Tuning (MRT), a novel approach that focuses on directly editing semantically rich multimodal representations to achieve strong performance and provide intuitive control over LMMs. Empirical results show that our method surpasses current state-of-the-art baselines with significant performance gains (*e.g.*, 1580.40 MME score) while requiring substantially fewer tunable parameters (*e.g.*, 0.03% parameters). Additionally, we conduct experiments on editing instrumental tokens within multimodal representations, demonstrating that direct manipulation of these representations enables simple yet effective control over network behavior.

## 1 INTRODUCTION

In this transformative era, artificial intelligence is undergoing a groundbreaking revolution, driven by the rapid rise of Large Multimodal Models (LMMs) (Dumas et al., 2009; Alayrac et al., 2022; Yin et al., 2023; Khattak et al., 2023). These models have demonstrated impressive capabilities across various multimodal tasks, spanning remarkable capacities in natural language processing, computer vision, and beyond. Imagining future development, a key objective in advancing LMMs is enhancing their zero-shot generalization ability to novel multimodal tasks. In this pursuit, multimodal instruction tuning has been introduced (Liu et al., 2024), full fine-tuning pre-trained models with diverse multimodal instruction-following datasets, thereby enabling zero-shot generalization to previously unseen multimodal tasks.

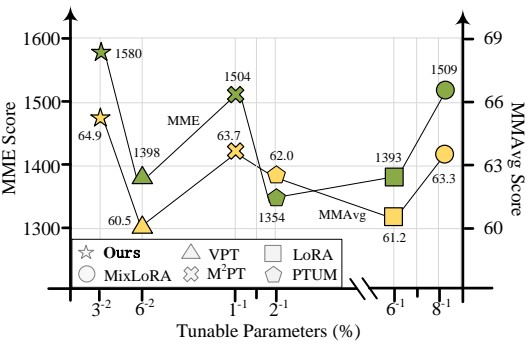

Figure 1: **MRT (ours)** *v.s.* **concurrent arts.** Our method yields significant performance gains over state-of-the-art multimodal PEFT approaches on MME and MMAvg benchmarks with considerably lower parameter usage (see Table 1).

However, LMMs continue to grow in parameter size and complexity (*e.g.*, LLaVA (Liu et al., 2024) leverages 7B and 13B backbone LLMs and Flamingo (Alayrac et al., 2022) employs 70B LLM). The standard approach of full fine-tuning LMMs from scratch presents significant challenges, as

---

[*]Equal contribution

[†]Corresponding author

researchers encounter difficulties in fine-tuning these pre-trained models both effectively and efficiently. A promising solution, similar to vision and language domains, is to utilize Parameter-Efficient Fine-Tuning (PEFT) strategies (Han et al., 2023; 2024b; Shen et al., 2024). Despite achieving promising effectiveness and efficiency, there are two main limitations in existing parameter-efficient methods. **First**, they typically require a substantial number of parameters to attain sub-par performance to full fine-tuning. Meanwhile, the potential of fine-tuning rich semantic multimodal representations has been largely overlooked; **Second**, The parameters introduced in the PEFT procedure are abstract and independent of the physical characteristics of the problem being modeled (Angelov & Soares, 2020). Consequently, they are challenging to interpret in a manner that aligns with human understanding (Li et al., 2018b; Jin et al., 2024b;c).

This perspective raises two key questions: ❶ *How can we achieve the **effectiveness** and **efficiency** of fine-tuning large-scale multimodal models?* ❷ *How can we explore the **controllability** of PEFT methods?* These two questions form the foundation of our work. Our intuition is that instead of merely modifying parameters in a black-box manner, as has been done in previous PEFT methods, we should explicitly investigate the potential of linearly interpretable representation engineering during the multimodal fine-tuning process. By doing so, we can not only improve the parameter efficiency but also foster a deeper understanding of the model's behavior, paving the way for advanced LMM efficiency and controllability.

In response to question ❶, we propose an efficient and effective representation fine-tuning strategy — Multimodal Representation Tuning (MRT), to explore the extreme of tunable parameters (*e.g.*, up to 21 times fewer parameters compared to LoRA) while achieving superior performance (*e.g.*, verses 4.7% higher performance on the MME benchmark (Fu et al., 2023b) compared to the state-of-the-art baseline MixLoRA (Shen et al., 2024)) (see Figure 1). To the best of our knowledge, MRT is the first work studying parameter-efficient multimodal representation tuning, inspired by the current representation fine-tuning for language models (Wu et al., 2024a;b; Turner et al., 2023).

To address question ❷, we demonstrate that directly editing multimodal representations can effectively control model behavior (see §3.3). Moreover, our findings indicate that precise behavior control offers valuable insights into the transparency and interpretability of PEFT methods, a topic that has been largely underexplored. We believe these insights establish foundational setup and perspectives for future research on multimodal representation understanding.

## 2 RELATED WORK

**Multimodal Instruction Tuning.** Transformers-based architectures currently dominate in LMMs, enabling breakthroughs in tasks such as visual question answering (Hu et al., 2024; Antol et al., 2015; Guo et al., 2023), image captioning (Özdemir & Akagündüz, 2024), and visual commonsense reasoning (Chen et al., 2024; Park et al., 2024). A general structure of LMMs includes three main components (Liu et al., 2024; Li et al., 2023b): a pre-trained modality encoder to encode modal features, a pre-trained LLM to reason fused multimodal data and perform prediction, and a cross-modality layer to align different modalities (*e.g.*, a linear projector in LLaVA (Liu et al., 2024) and MiniGPT4 (Zhu et al., 2024), a GATED XATTN-DENSE layer in Flamingo (Alayrac et al., 2022)). An effective tuning method in improving the zero-shot capability of LMMs is multimodal instruction tuning (Liu et al., 2024; Zhu et al., 2024; Dai et al., 2023). It refines LMMs by fine-tuning diverse instruction-following datasets that embrace both user intent and desired responses, including machine-generated and human-annotated data. In this work, we explore parameter-efficient multimodal instruction tuning on LLaVA.

**Parameter-Efficient Fine-Tuning.** Parameter-Efficient Fine-Tuning (PEFT) has emerged to solve the computational challenges of adapting large-scale models (*e.g.*, LLMs, LMMs) to downstream tasks (Wang et al., 2024; Liu et al., 2024), aiming to achieve comparable performance to full fine-tuning while updating only a small fraction of model parameters or training customized learnable modules. Current PEFT strategies can be generally categorized into three groups: *reparameterization*, *layer insertion* and *prompt tuning*. *Reparameterization* methods (*e.g.*, LoRA (Hu et al., 2022), IA3 (Liu et al., 2022)) mainly focus on the reparameterization of the attention mechanism, offering a balance between efficiency and performance. However, these methods still require a great amount of parameters while leaving a noticeable performance gap compared to full fine-tuning. *Layer Insertion* methods (*e.g.*, Adapters (Long et al., 2024)) generally insert learnable modules (*e.g.*, fully-connected layers) between attention or MLP. Nevertheless, they typically have higher parameter usage and ad-

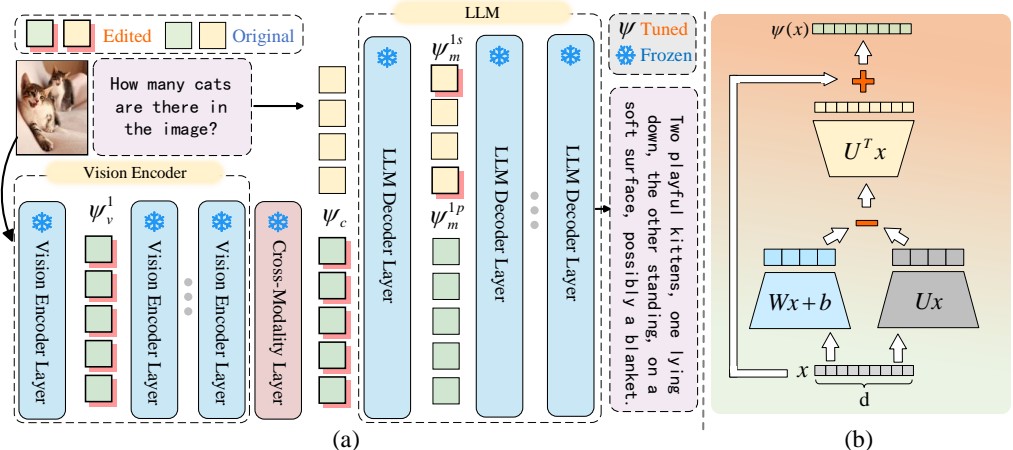

Figure 2: **Overview of MRT.** Representation editors $\psi \in \{\psi_V, \psi_c, \psi_P, \psi_S\}$ are the only tunable parameters while the entire model remains completely frozen. During fine-tuning, we jointly edit the visual representations in the vision encoder, the cross-modality layer, and the prefix and suffix of textual-oriented fraction in the multimodal representations in the LLM. These editors efficiently and effectively optimize the model representations during multimodal instruction tuning.

ditional burden during inference. *Prompt-tuning* (Jia et al., 2022; Yan et al., 2023) adds learnable soft tokens as a prefix to guide pre-trained models for specific tasks. While prompt tuning is more parameter-efficient than *Reparameterization* and *Layer Insertion* methods, training only the prompt embedding could lead to sub-optimal performance when encountering more complicated tasks.

From a different perspective, recent advances in representation engineering (Turner et al., 2023; Zou et al., 2023; Geiger et al., 2021) raise the exploration into representation tuning in Nature Language Processing (NLP) and Computer Vision (CV) fields, demonstrating promising results and superior parameter efficiency in comparison to existing PEFT methods. Specifically, RED (Wu et al., 2024a) proposes a direct representation editing, utilizing element-wise scaling and a bias for the entire representation of Transformers-based layers. ReFT (Wu et al., 2024b) introduces intervention-based representation editing, steering partial representations of Transformers-based layers via a low-rank projection matrix with orthonormal rows and a linear projector. Although representation tuning has shown its exceptional capability on single modality (*i.e.*, language), its effectiveness on multi-modalities is largely unexplored. Our method is the pioneering work to investigate the feasibility of multimodal intervention-based representation tuning via rigorous structural design. Additionally, our experiments on instrumental token editing demonstrate that modifications within multimodal representations are highly effective, enabling precise counterfactual control over network behavior in the multimodal PEFT approach — an area that has not yet been sufficiently explored.

## 3 METHODOLOGY

In this section, we introduce MRT, a pioneer multimodal representation tuning approach for effective and efficient LMM fine-tuning. We first introduce the preliminary of LMMs and notations in §3.1. The effective representation tuning with the designing of editors in visual, cross-modality, and multimodal representation are presented in §3.2. The overall framework is shown in Figure 2.

### 3.1 PRELIMINARY

Given a vision-text Transformers-based LMM $\mathcal{F}$, which has been pre-trained on a substantial corpus of data and tasks, the model architecture is composed of three major components: a vision encoder $\mathcal{V}$ with hidden dimensionality $d_v$, a large language model $\mathcal{T}$ with hidden dimensionality $d_t$ and a linear cross-modality projector $\mathcal{C}$ that aligns the dimensionality of visual features from $d_v$ to $d_t$. The input of the model $\mathcal{F}$ is an image $\mathtt{I}$ and text instruction $\mathtt{T}$.

The processing of these inputs proceeds through the following steps: firstly, the vision encoder $\mathcal{V}$ transforms the image $I$ into a $d_v$-dimensional visual tokens, denoted as $T_v = \mathcal{V}(\mathtt{I}) = \{T_v^1, T_v^2, \ldots, T_v^m\}$, where $m$ is the number of visual tokens generated by the encoder. Sec-

ondly, the cross-modality projector $\mathcal{C}$ maps the visual representation $I$ to the dimensionality of the language model, producing a $d_t$-dimensional visual embedding $X_v = \mathcal{C}(T_v) = \{\mathcal{C}(T_v^1), \mathcal{C}(T_v^2), \ldots, \mathcal{C}(T_v^m)\} = \{x_v^1, x_v^2, \ldots, x_v^m\}$. Parallelly, the text instruction $T$ is tokenized into a sequence of textual tokens, $X_t = tokenize(\text{T}) = \{x_t^1, x_t^2, \ldots, x_t^n\}$, where $n$ represents the number of tokens with $d_t$-dimensional space, forming a textual embedding from the text instruction. Lastly, the visual representation $X_v$ and the textual representation $X_t$ are combined through a fusion mechanism, yielding a joint multimodal representation $X = \texttt{Concat}(X_v, X_t)$. Based on this fused representation, the large language model $\mathcal{T}$ generates a relevant linguistic response $y = \mathcal{F}(\text{I}, \text{T}) = \mathcal{T}(\texttt{Concat}(X_v, X_t))$.

In our study, the primary objective is to fine-tune the pre-trained model, to enhance its zero-shot performance. While prior research has explored full fine-tuning strategies as well as parameter-efficient fine-tuning methods (see §2), we propose Multimodal Representation Tuning (MRT) that offers a more computationally efficient approach. MRT has the potential to enhance performance significantly while minimizing resource consumption (see §4.2), presenting an advantageous alternative to existing fine-tuning techniques from a representation view.

## 3.2 MULTIMODAL REPRESENTATION TUNING

MRT is inspired by the linear representation hypothesis (Wu et al., 2024b) and interchange interventions (Geiger et al., 2021). As shown in Figure 2, we apply representation editors for each layer of the vision encoder, LLM, and cross-modality layer, optimizing visual representation, cross-modality representation, and multimodal representation simultaneously. Notably, during multimodal instruction tuning, MRT only updates these editors, while the entire model remains completely frozen.

**Representation Editor.** We introduce a representation editor $\psi$ formulated via the simple yet effective representation hypothesis. The editor modifies the original feature representation $x$ within a specific subspace to reflect the desired intervention obtained from a linear projection $Wx + b$, where both $W$ and $b$ are learnable parameters. The editing operation is then confined to the subspace spanned by the rows of a low-rank matrix $U$ with orthonormal rows, so only targeted aspects of the representation are adjusted while preserving the remaining information. The editor $\psi$ is:

$$\psi(x) = x + U^{\intercal}(Wx + b - Ux), \tag{1}$$

where $U$ and $W \in \mathbb{R}^{d_l \times d_t}$ are low-rank matrices (*i.e.*, $d_l \ll d_t$); $d_l$ represents the rank of the subspace. $U^{\intercal}$ denotes the transpose of $U$. Specifically, $Ux$ projects the original representation onto the subspace $U$, where $Wx + b$ provides the target values within that subspace via linear transformation from the original feature representation $x$. The difference $Wx + b - Ux$ computes the necessary interventions, which are then mapped back into the original space via $U^{\intercal}(\cdot)$. By adding this intervention to the originals, we obtain the edited representation $\psi(x)$ that incorporates the desired modifications while maintaining the components orthogonal to the subspace $U$.

This editor facilitates controlled manipulation of representations by targeting linear subspaces, as multiple studies have shown that human-interpretable concepts are encoded linearly (Smolensky, 1986; Rumelhart et al., 1986; Lasri et al., 2022; Guerner et al., 2023; Mikolov et al., 2013). Consequently, linear subspace interventions can correspond to specific semantic attributes and modalities, thereby advancing research in LMM interpretability and controllability (see §3.3). Utilizing a low-rank subspace with orthonormal rows not only enhances computational efficiency but also contributes to the stability and effectiveness of the intervention process.

**Visual Representation.** Without loss of generality, we maintain consistency with the previously established notation and recall the vision encoder $\mathcal{V}$, which extracts visual features from a given image $I$. For visual representation intervention, we define a set of visual representation editors, denoted as $\psi_V = \{\psi_v^1, \psi_v^2, \ldots, \psi_v^i\}$, where each individual editor $\psi_v^i$ corresponds to a distinct visual low-rank representation editor that operates at the $i$-th layer of the encoder $\mathcal{V}$. These editors function to modify the hidden visual representations $T_{v,i}$ at their respective layers within the encoder.

Specifically, each editor $\psi_v^i$ edits the complete set of hidden visual representations produced at its corresponding layer, expressed as:

$$T_{v,1} = \{\psi_v^1(T_{v,1}^1, T_{v,1}^2, \cdots, T_{v,1}^m)\},$$
$$\vdots \qquad\qquad (2)$$
$$T_{v,i} = \{\psi_v^i(T_{v,i}^1, T_{v,i}^2, \cdots, T_{v,i}^m)\},$$

where $T_v^i$ represents the set of visual representations at $i$-th layer. They are sequentially edited by the corresponding editor $\psi_v^i$. The process is applied across all $m$ hidden representations at each layer, ensuring that each layer's output receives a layer-specific intervention. Finally, the edited visual representation from the second last layer is fed into the cross-modal projection layer $\mathcal{C}$ (See §S2).

**Cross-modality Representation.** In multimodal models, aligning visual representations within a unified representation space is crucial. The cross-modality projector $\mathcal{C}$ plays an important role in this process. Consequently, we introduce an intervention in the cross-modality projector $\mathcal{C}$, aiming to improve the alignment between visual and textual features.

Specifically, we define a cross-modality editor, denoted as $\psi_c$. The visual features $X_v$ are first processed by the cross-modality projector $\mathcal{C}$, which integrates representations from each layer of the visual encoder $\mathcal{V}$. The editor $\psi_c$ is then applied to the output of $\mathcal{C}$, yielding an optimized visual representation $X_v$, defined as:

$$X_v = \psi_c(\mathcal{C}(\{T_v^1, T_v^2, \cdots, T_v^m\})), \qquad (3)$$

where $T_v$ represents the hidden visual representations from the final layer of the encoder $\mathcal{V}$. This edited visual representation, $X_v$, is subsequently combined with the corresponding textual representation $H_t$, producing a unified multimodal representation. The refined visual representation $X_v$, incorporating the output of the cross-modality editor $\psi_c$, is concatenated with the textual representation $X_t$, ensuring effective alignment of both modalities.

**Multimodal Representation.** In the previous discussion, the visual representation has been comprehensively processed through $\psi_v$ and $\psi_c$. In this part, we shift toward the textual-oriented embedding tokens within the multimodal representations, where the visual and textual embeddings are concatenated together (see §3.1).

We concentrate on editing solely the textual-oriented representations, as the image representations have already been extensively modified through $\psi_V$ and $\psi_c$. We manipulate two consecutive segments of the textual embeddings, corresponding to $a$ prefix tokens and $b$ suffix tokens. This process is facilitated by two sets of multimodal representation editors: $\psi_p = \{\psi_p^1, \psi_p^2, \ldots, \psi_p^j\}$ for the prefix tokens and $\psi_s = \{\psi_s^1, \psi_s^2, \ldots, \psi_s^j\}$ for the suffix tokens. Here, $\psi_p^i$ and $\psi_s^i$ denote the low-rank multimodal representation editors responsible for modifying the textual-oriented prefix and suffix embeddings, respectively, at the $i$-th layer of the textual encoder $\mathcal{T}$. Formally, the process of editing the multimodal representations across the $j$ layers is defined as:

$$X_1 = \{x_{v,1}^1, \cdots, x_{v,1}^m, \psi_p^1(x_{t,1}^1, \cdots, x_{t,1}^a), x_{t,1}^{a+1}, \cdots, x_{t,1}^{n-b-1}, \psi_s^1(x_{t,1}^{n-b}, \cdots, x_{t,1}^n)\},$$
$$\vdots \qquad\qquad (4)$$
$$X_j = \{x_{v,j}^1, \cdots, x_{v,j}^m, \psi_p^j(x_{t,j}^1, \cdots, x_{t,j}^a), x_{t,j}^{a+1}, \cdots, x_{t,j}^{n-b-1}, \psi_s^j(x_{t,j}^{n-b}, \cdots, x_{t,j}^n)\},$$

where $x_{v,j}^1, \ldots, x_{v,j}^m$ represent the visual tokens at $j$-th layer, and $x_{t,j}^1, \ldots, x_{t,j}^n$ represent the textual tokens. The prefix editors $\psi_p^j$ and suffix editors $\psi_s^j$ apply the targeted intervention to the $a$ prefix and $b$ suffix textual tokens, following common practice (Geiger et al., 2021; Wu et al., 2024b). Altogether, by concatenating edited visual and textual tokens, we are able to adjust the intricate relationships between visual and textual information across layers. Further, as the visual and textual editing are decoupled, we are then able to facilitate accurate LMM controllability (see §3.3).

### 3.3 CONTROLLABILITY: THE BUTTERFLY EFFECT.

Controllable Text Generation (CTG) has been a recent surge of interest in the field of NLP for high-quality or task-oriented generation, covering several conditions related to lexical, structural,

and semantic aspects (Zhang et al., 2022; Khalifa et al., 2020; Erdem et al., 2022). However, many efforts (Zhang et al., 2023; Zeldes et al., 2020; Gao et al., 2020) designed to control the model in an implicit way to drive the generation of text satisfying specific conditions; the transparency and simplicity of CTG, however, remain problematic and misleading (Rudin, 2019; Rudin et al., 2022; Laugel et al., 2019; Arrieta et al., 2020). Existing methods for generation control can be broadly categorized into *post-processing* and *model behavior adjustment* (Zhang et al., 2023; Liang et al., 2024; Petrov et al., 2023). *Post-processing* re-ranks the original next-token distributions in the textual decoder as a filter to control the desired type of text while keeping the model completely frozen. Though intuitive, it remains challenging to achieve better control performance. Even worse, in the multimodal scenario, the visual representation becomes entirely uncontrollable due to its decoder-oriented design. *Model behavior adjustment* utilizes strategies such as full fine-tuning, prompt tuning, and adapter to satisfy the controlled conditions. While effective, the behavior adjustment remains implicit, relying solely on full or partial parameter updates. The semantic meanings of representations within models, however, have largely gone unexplored.

In light of this view, we investigate the LMM controllability from a representation perspective, aiming to edit the actual semantics directly in a flexible and explicit manner. We draw upon current research in LLM interpretability (Geiger et al., 2021; Wu et al., 2024b), where training a set of low-rank causal interventions on selected residual streams can effectively induce a base LLM to follow human-desired instructions. Namely, given a singular set of representations, our design is able to manipulate them in a targeted manner to achieve generalized control. In §4.3, we demonstrate that, even within complex multimodal settings, it remains feasible to interpret individual neurons and representations in isolation. We believe that this represents a significant advancement towards multimodal interpretability and controllability.

## 4 EXPERIMENT

### 4.1 EXPERIMENTAL SETUP

**Implementation details.** Following common practice (Liu et al., 2024; Wang et al., 2024), we employ stage-one LLaVA (Liu et al., 2024) with CLIP-L (*i.e.*, 24 Transformers-based encoder layers) as the vision encoder, a pre-trained cross-modality projector and Vicuna-7B-v1.3 (Chiang et al., 2023) (*i.e.*, 32 Transformers-based decoder layers) as the backbone LLM in our pre-trained LMM (see §3.1). For both visual representation editing and multimodal representation editing, we implement the same editor structure (see §3.2). For visual representations, we edit the entire visual representation in CLIP-L and the cross-modality layer. For multimodal representations, we apply editing for both textual-oriented prefixes and suffixes in Vicuna-7B-v1.3. More implementation details and discussion on inference time are included in Appendix §S2.

**Datasets**. We conduct multimodal instruction tuning on Vision-Flan (Xu et al., 2024), a human-annotated multimodal instruction tuning dataset with 191 diverse tasks. Following common practice (Shen et al., 2024), we employ the scaled-down version containing up to $1,000$ instances per task, resulting in a total of $191,105$ instances. For evaluation, we examine our method on the multimodal evaluation benchmark MME (Fu et al., 2023a), measuring both perception and cognition abilities across 14 subtasks (see §S1). We further investigate the model's capabilities using 7 multimodal datasets. Specifically, we utilize the Text-VQA (Singh et al., 2019) for Optical Character Recognition, and the Visual Spatial Reasoning (VSR) (Liu et al., 2023) for reasoning. The perception capability is tested on CIFAR-10/100 (Krizhevsky et al., 2009) and MNIST (Deng, 2012). Moreover, the SNLI-VE dataset (Xie et al., 2019) evaluates Visual Entailment capabilities, while the POPE (Li et al., 2023c) dataset examines the object hallucination tendencies.

**Evaluation Metrics.** The MME incorporates both Perception and Cognition metrics[1] for evaluation. For other multimodal datasets, we use Vicuna-13B-v1.5 (Zheng et al., 2024) to assess the accuracy of each prediction compared to the groundtruth, as suggested by common practice (Shen et al., 2024; Wang et al., 2024; Han et al., 2024b).

### 4.2 MAIN RESULTS

In Table 1, we report a comprehensive zero-shot evaluation of MRT on eight multimodal datasets, comparing with several baselines. Specifically, we consider seven state-of-the-art PEFT methods, in-

---

[1]https://github.com/BradyFU/Awesome-Multimodal-Large-Language-Models/tree/Evaluation

Table 1: **Zero-shot Multimodal Evaluation.** LLaVA$_{Align}$ is the stage-one LLaVA without end-to-end fine-tuning, and LLaVA$_{FT}$ indicates the fully fine-tuned LLaVA. The MMAvg represents the average score on the right seven tasks. Vision-Flan dataset is used for all fine-tuning processes. The best performance except LLaVA$_{FT}$ is shown in **bold**, and the second best is shown in underline. As seen, MRT outperforms current state-of-the-art methods with far fewer trainable parameters (*i.e.*, 0.03%). Note that the we have calibrated the M$^2$PT's # para as it should include both 0.09% soft prompt params, and 1.87% lm head params for the overall parameter usage.

| Method | # para | MME | Text-VQA | VSR | SNLI-VE | CIFAR-10 | CIFAR-100 | MNIST | POPE | MMAvg |
|---|---|---|---|---|---|---|---|---|---|---|
| LLaVA$_{Align}$ (Liu et al., 2024) | - | 1110.82 | 32.62 | 50.16 | 34.51 | 80.00 | 58.04 | 52.79 | 59.10 | 52.46 |
| LLaVA$_{FT}$ (Liu et al., 2024) | 100% | 1587.26 | 37.26 | 53.76 | 43.35 | 92.97 | 63.73 | 94.27 | 80.82 | 66.59 |
| LoRA (Hu et al., 2022) | 0.63% | 1393.67 | 39.20 | 52.95 | 44.56 | 90.10 | 45.90 | 83.42 | 72.33 | 61.21 |
| APrompt (Wang et al., 2023a) | 0.23% | 1406.63 | 35.26 | 53.12 | **45.58** | 85.74 | 50.27 | 84.63 | 76.16 | 61.52 |
| PTUM (Yang et al., 2023) | 0.12% | 1354.62 | 34.28 | 53.75 | 30.86 | 82.88 | 57.63 | 94.29 | 80.31 | 62.00 |
| VPT (Han et al., 2024b) | 0.06% | 1398.74 | 33.68 | **53.93** | 32.62 | 76.49 | 52.31 | 94.73 | 79.60 | 60.48 |
| ReFT (Wu et al., 2024b) | 0.03% | 1473.25 | 36.34 | 49.75 | 39.66 | 90.43 | 57.53 | 88.21 | 78.35 | 62.90 |
| M$^2$PT (Wang et al., 2024) | 1.96% | 1503.98 | 34.48 | 53.19 | 32.89 | 89.29 | 59.14 | 95.54 | **81.26** | 63.68 |
| MixLoRA (Shen et al., 2024) | 0.85% | 1509.61 | 40.42 | 49.18 | 36.69 | 91.40 | 59.27 | 87.68 | 78.48 | 63.30 |
| MRT | 0.03% | **1580.40** | **40.62** | 51.47 | 33.34 | **96.96** | 57.20 | **95.63** | 79.30 | **64.93** |

cluding LoRA (Hu et al., 2022), APrompt (Wang et al., 2023a), PTUM (Yang et al., 2023), VPT (Han et al., 2024b), M$^2$PT (Wang et al., 2024), MixLoRA (Shen et al., 2024) and ReFT (Wu et al., 2024b). Here LoRA and MixLoRA are reparameterized methods, initializing and updating extra low-rank decomposition matrices within attention blocks; APrompt, VPT, PTUM, and M$^2$PT are prompt tuning methods. Differently, APrompt and VPT consider only inserting tunable soft prompts to a single modality (*i.e.*, textual and visual space, respectively) while PTUM and M$^2$PT are multi-modal prompt tuning approaches; ReFT is the most recent representation tuning approach for textural modality. We do not include layer insertion methods in this comparison, as they typically require significantly higher parameter usage (Wu et al., 2024b; Balne et al., 2024), rendering them unsuitable under the multimodal PEFT settings. Consequently, we have several key observations. *First*, MRT **outperforms all** PEFT methods with substantial performance gains. For example, our approach achieves **4.70%** and **5.08%** improvements on MME compared to two state-of-the-art PEFT baselines, MixLoRA and M$^2$PT, respectively. MRT can be further considered as a qualified alternative to full fine-tuning, as it reaches **99.56%** of the overall full fine-tuning performance on MME while introducing only **0.03%** of the model parameters, demonstrating both its effectiveness and efficiency for large-scale multimodal model adaptation. Diving into the per-task performance, we also want to highlight that MRT outperforms the full fine-tuning LLaVA on Text-VQA, CIFAR-10, and MNIST tasks with a large performance gap (*i.e.*, 3.36%, 5.99%, 3.36%). *Second*, we observe that PEFT approaches focusing on multimodality (*i.e.*, M$^2$PT, MRT) generally outperform other methods that consider only a single modality (*i.e.*, APrompt, VPT, ReFT, MixLoRA). This indicates the significance of introducing cross-modality interactions within MRT. The ablation study on component ablation in §4.4 further proves that exploiting multimodal representation editing can result in higher performance. *Third*, similar to other PEFT approaches (*e.g.*, PTUM, M$^2$PT), MRT does not perform very well on visual entailment task, SNLI-VE. We hypothesize it's due to the complexity of logical relationship understanding, which might require a more sophisticated task-oriented design.

## 4.3 Controllability Results

We design our experiment on several image classification tasks, where we take an image-question pair as inputs. The LMM further answers the question based on the class prediction. As discussed in §3.3, our objective is to design targeted representation tuning that effectively intervenes in a few selected instrumental visual-based (*i.e.*, visual and cross-modality tokens), and multimodal tokens to generate semantically counterfactual outputs.

Specifically, regarding that both visual-based features and textual-oriented target indicators in multimodal representations are rich in semantic information and play crucial roles in the image classification task (Parekh et al., 2024), we decouple and study the LMM controllability via a set of representation editors $\psi = \{\psi_v^1, \psi_c^1, \psi_t^1\}$ from both modalities. Here $\psi_t^1$ indicates the multimodal representation editor for the targeted textual-oriented token at the first layer of the LLM (*i.e.*, not $\psi_p$ or $\psi_s$, but rather the specific token position we intend to control).

Shown in Figure 3, for visual-based representation (*i.e.*, $T_v, X_v$) editing, given that all images are represented as visual-based token patches of fixed length, our analysis here concentrates on the

semantically salient Regions of Interest (RoI), specifically the most informative visual-based patches (*e.g.*, objects for image classification). Note that we consider only RoIs as candidates in this setting, which is different from §3.2. The reason is that, during instruction fine-tuning, it is essential to consider all visual-based tokens for effective feature editing, whereas in targeted semantic control, only the RoIs align the most to the paired question. Thus, editors $\psi_v^1$ and $\psi_c$ are trained to edit only RoIs to control the most important semantic information.

For editing of textual-oriented target indicators in multimodal representations, we control the textual questions to a fixed template (More templates are shown in Appendix §S6): *"Is the object an [indicator] in the image?"* The representation editor $\psi_t^1$ is trained to modify only the token corresponding to *"[indicator]"* within this sequence (*i.e.*, the 5-th token). Given an image of class $e$, and the question *"Is the object an e in the image?"*, an affirmative response (*i.e.*, *"Yes"*) represents a correct classification, while a negative response (*i.e.*, *"No"*) refers to the incorrect one.

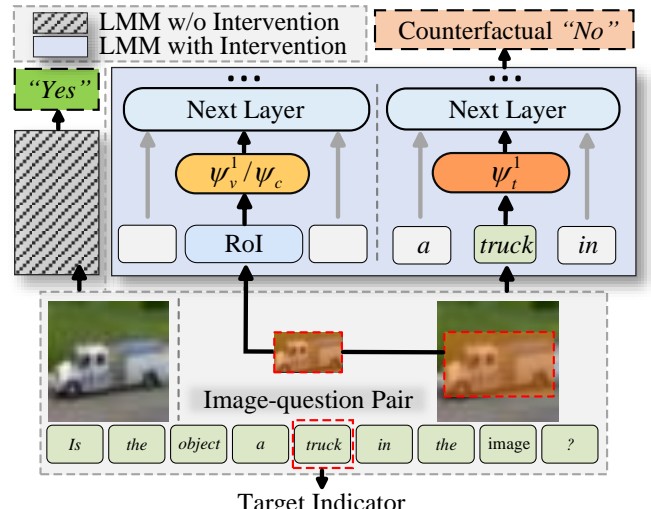

Figure 3: **Controllabilty Pipeline on Image Classification.** MRT offers LMM controllability from a representation perspective, allowing for direct editing of representations with semantic meanings and enabling counterfactual interference with the results. Details are shown in §4.3.

Specifically, we target two different scenarios of counterfactual outputs: **i) Misclassification.** Counterfactual output of misclassification on a specific class $e$ while achieving high classification accuracy for other classes. For training editors $\psi_1$, we change the training data where all labels of the targeted class $e$ to counterfactual "No" while keeping the groundtruth labels "Yes" of other classes. **ii) Misalignment.** Counterfactual output of misaligning a specific class $e$ into another class $\bar{e}$. We train our editor $\psi_2$ with misaligned image class $e$ with groundtruth of $\bar{e}$. Note that $\psi_1$ and $\psi_1$ are two independent sets for each scenario.

In Table 2, we conduct and report the results of control over counterfactual output on 5 randomly selected classes from CIFAR-10 (Krizhevsky et al., 2009) (*i.e.*, cat, dog, ship, frog and truck). For **i)**, the trained representation editors $\psi$ can modify the representation to produce a counterfactual re-

Table 2: **Controlled Counterfact Rate** is evaluated on two scenarios: misclassification and misalignment.

| Class $e$ ($LLaVA_{Align}$) | Misclassification | | Misalignment | |
|---|---|---|---|---|
| | Misclassification on $e$ | *Others* | Misalignment to $\bar{e}$ | *Others* |
| (a) cat   18.8% | 100% | 0% | 100% | 0% |
| (b) dog   17.3% | 100% | 0% | 100% | 0% |
| (c) ship   21.8% | 100% | 0% | 100% | 0% |
| (d) frog   22.5% | 100% | 0% | 100% | 0% |
| (e) truck   21.4% | 100% | 0% | 100% | 0% |

sponse (*i.e.*, from *"Yes"* to *"No"*) with 100% success rate, effectively causing the model to misclassify all class $e$ images when presented with the same template question prompts. Notably, interfering with class $e$ does not prevent the model from classifying other classes correctly. For **ii)**, the result demonstrates that the representation editors can fully control (*i.e.*, 100%) the model to misalign all images from class $e$ into the targeted class $\bar{e}$, while maintaining the capability of accurate classification of other classes. All together, our results clearly show that by simple yet intuitive token-wise representation editing, one can directly control the complex decision-making process, even considering multimodal information as inputs. Furthermore, our insights may significantly advance LMM interpretability research, as the editing process is directly applied to both visual and multimodal representations, thereby regularizing trackable representations with certain properties (*i.e.*, in our case, linear projection). By further designing explicitly the casual model, MRT can show potential as a promising solution for achieving *ad-hoc* interpretability (Wang et al., 2023b; Subramanian et al., 2018; Chen et al., 2016; Jin et al., 2024a; Han et al., 2024a).

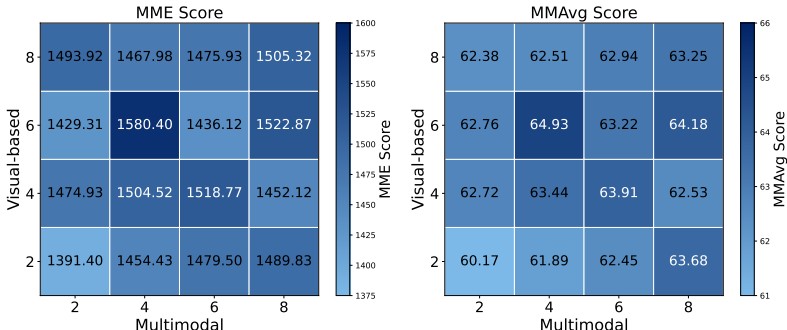

Figure 4: **Impact of Rank.** Each cell in the map corresponds to the evaluation score of a model with a multimodal rank (row) and a visual rank (column). A darker hue represents a higher score, whereas a lighter hue indicates a lower score.

## 4.4 Diagnostic Experiments

**Impact of Rank.** In MRT, the number of ranks directly determines the number of tunable parameters. To further analyze the impact of rank $w.r.t.$ model performance, we conduct a comprehensive study on the number of ranks of visual-based and multimodal editors on Vision-Flan. Specifically, we use grid search to consider different combinations of ranks for visual-based, and multimodal editors, ranging from rank 2 to 8, respectively. We propose altering the visual and cross-modality editors to maintain the same rank count, eliminating the need for an additional manually set value, in alignment with §4.3. The results are reported in Figure 4. As seen, the optimal configuration, yielding a peak score of 1580.40, is achieved with visual-based rank 6, and multimodal rank 4. We stop at rank 8 because performance saturation is observed around this point. Further increasing the rank would result in increased parameter usage without significant performance improvement (*e.g.*, 1505.32 on MME with visual-based and multimodal rank 8, and 1452.12 with visual-based rank 4 and multimodal rank 8). This may result from slower convergence or overparameterization (Han et al., 2023; Hu et al., 2022; Shen et al., 2024; Zeng et al., 2024).

**Discussion on Optimization.** We further investigate why MRT exhibits superior performance and generalizes effectively across different tasks from an optimization perspective. Previous studies (Li et al., 2018a; Ma et al., 2022) have shown that the geometry of the loss landscape plays a crucial role in model generalization. Building on this insight, we depict the loss landscape in Figure 5. Here, we randomly choose two parameter directions, as the choice of random directions has been shown to have minimal impact on the results (Li et al., 2018a). As seen, MRT provides a larger connected region around the local minimum (*e.g.*, the yellow square area in the

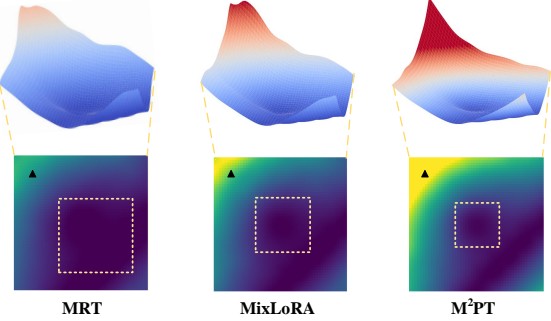

Figure 5: **Loss Landscape** along two random directions. The top three surfaces represent the loss landscape, while the bottom three are the 2-d heat maps.

heat map, where the larger dark blue area in MRT offers more optimization choices) and a smoother edge of the loss landscape for mitigating chaotic landscape (*e.g.*, ▲ in the heat map, where the sharpness in MixLoRA and M²PT is sensitive to loss fluctuations, leading to worse generality), indicating that MRT achieves a flatter loss landscape, which consistently corresponds with lower test error.

**Impact of Editing Position.** We further investigate the impact of editing positions in MRT (*i.e.*, visual, multimodal, and cross-modality representations) in Figure 6 left, removing each component individually from MRT's best rank combination to assess its contribution to the overall model performance. The results demonstrate that the model performance degrades when any tunable editors are excluded, which is consistent with our expectations. Moreover, we observe that the importance of different components is varied. For example, removing the cross-modality editor has the smallest

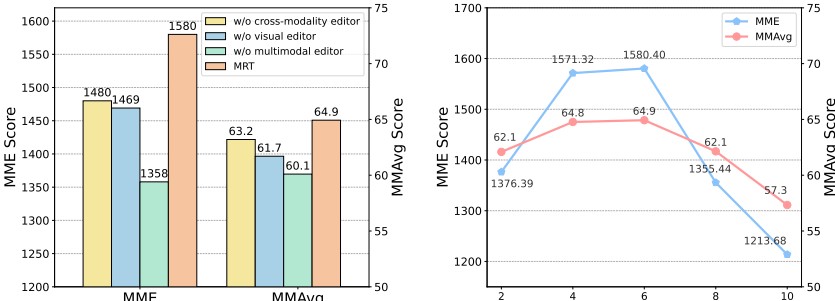

Figure 6: **Impact of Editing Position & Editing Length.** The left figure shows model performance under the different settings of representation editing position, while the right figure indicates the influence of different representation editing lengths.

impact on performance for both MME and MMAvg, while taking away either visual or multimodal editor leads to more significant performance drops.

**Impact of Editing Length.** In Figure 6 right, we explore representation editing length. We focus on the variance of textual-oriented representation lengths, as visual representations generally lack the semantic segmentability characteristic (*i.e.*, it is ineffective to include only partial visual patches for editing, as image features are typically encoded and evaluated from a holistic perspective). We thus do not change the editing of visual representation as mentioned in §3.2. By extending the range of both prefixes and suffixes length from 2 to 10, our findings reveal a non-linear relationship between intervention length and performance efficacy. Specifically, we observed optimal results with editing lengths of 4 and 6 for both prefixes and suffixes (*i.e.*, 1580.40 and 1571.32 on MME), while the trend on MMAvg is also consistent with this observation. Shorter lengths (*e.g.*, 2) appear to be insufficient to capture the necessary contextual information or to adequately modify the representation. Conversely, longer lengths (*e.g.*, 8, 10) result in slower convergence or over-interference, potentially over-disrupting the pre-trained LMM.

**Impact of Editing Depth.** Following common practice (Han et al., 2024b; Wang et al., 2024; Jia et al., 2022), we examine the influence of editing depth for visual and multimodal representation editors to the overall model performance under 5 different settings: (a) the first layer; (b) every even-number layer (*i.e.*, $i \in [2, 4, \ldots, 22]$, $j \in [2, 4, \ldots, 32]$); (c) the first half of the layers (*i.e.*, $i \in [1, 2, \ldots, 12]$, $j \in [1, 2, \ldots, 16]$); (d) the latter half of the layers (*i.e.*, $i \in [12, 13, \ldots, 23]$, $j \in [16, 17, \ldots, 32]$); and (e) all layers. Each setting reports the best rank

Table 3: **Impact of Editing Depth.**

| Editing Depth | MME | MMAvg |
|---|---|---|
| (a) First Layer | 1329.84 | 60.57 |
| (b) Odd Layers | 1468.21 | 63.35 |
| (c) First Half | 1440.32 | 61.89 |
| (d) Latter Half | 1447.41 | 62.65 |
| (e) All Layers | **1580.40** | **64.93** |

combination selected by MME. As seen in Table 3, MRT's performance is positively correlated with editing depth. Additionally, we find that even with minimal editing depth (*i.e.*, setting (a)), MRT demonstrates relatively strong performance, surpassing VPT on MMAvg (*i.e.*, 60.57 *v.s.* 60.48). Editing only the latter half of the layers yields better performance compared to editing the first half (*i.e.*, 1447.41 *v.s.* 1440.32 on MME). We also observe that editing at every odd layer outperforms both the "first half" and "latter half" configurations (*i.e.*, 1468.21 *v.s.* 1447.41 on MME), suggesting that distributing representation edits across the model in a sparse manner can be more beneficial than focusing on a continuous block of layers.

## 5 CONCLUSION

We introduce Multimodal Representation Tuning (MRT), an efficient and effective solution for parameter-efficient multimodal instruction tuning. It enjoys several advantages: **i)** MRT achieves remarkable parameter efficiency, utilizing up to 65 times fewer parameters than existing methods while achieving outstanding performance on multimodal evaluation benchmarks. It leverages the power of the semantically rich multimodal representations during PEFT, which have been largely overlooked in previous approaches; and **ii)** The accurate token-level multimodal representation control reveals the potential for enhanced controllability of multimodal models, paving the way for more transparent and interpretable text generation. As a whole, we conclude that the outcomes elucidated in this paper impart essential understandings and necessitate further exploration within this realm.

ETHICS STATEMENT

We conform to the ICLR Code of Ethics and further show the consent to our work below. All the datasets and benchmarks included in our study are publicly available (*i.e.*, Vision-Flan, MME, Text-VQA, Visual Spatial Reasoning (VSR), CIFAR-10/100, MNIST, SNLI-VE, POPE), and all the models are publicly available (see Appendix §S7 for Asset License and Consent). We would like to state that the contents in the dataset do NOT represent our views or opinions and our paper does not involve crowdsourcing or research with human subjects. More discussions are presented in Appendix §S10.

REPRODUCIBILITY STATEMENT

MRT is implemented in Pytorch (Paszke et al., 2019). Experiments are conducted on NVIDIA A100-40GB GPUs. Our full implementation is available at `https://github.com/comeandcode/MRT`. We include implementation details in §4.1 and Appendix §S2.

ACKNOWLEDGMENT

This research was supported by the National Science Foundation under No. 2242243, and the DEV-COM Army Research Laboratory under Contract W911QX-21-D-0001. The views and conclusions contained herein are those of the authors and should not be interpreted as necessarily representing the official policies or endorsements, either expressed or implied, of the U.S. DEVCOM Army Research Laboratory (ARL), U.S. Naval Research Laboratory (NRL) or the U.S. Government.

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

## SUMMARY OF THE APPENDIX

This supplementary contains additional details for the thirteenth International Conference on Learning Representations submission, titled *"Re-Imagining Multimodal Instruction Tuning: A Representation View"*. The supplementary is organized as follows:

- §S1 provides an additional **introduction of the datasets** used, including the number of examples and task categories.
- §S2 explains **more implementation details** on training and controllability experiments.
- §S3 presents the **evaluation metrics** used to assess the performance of the models.
- §S4 includes an **additional ablation study** on applying MRT to single modality.
- §S5 shows **a comparison of the inference time** across different models, emphasizing the inference efficiency of MRT.
- §S6 provides extended **controllability experiments and analysis**.
- §S7 presents related **asset license and consent** to our work.
- §S8 is the claim of **reproducibility**.
- §S9 discusses the **social impact and potential limitations** of our research.
- §S10 includes additional discussions on **ethics concerns**.
- §S11 reflects on the findings and provides **potential future directions** for improving and extending our work.

## S1 DATA STATISTICS

Details of 9 multimodal datasets for model instruction fine-tuning and multimodal evaluation are illustrated in Table S1. Vision-Flan (Xu et al., 2024) covers 191 distinct multimodal tasks which is ideal for our instruction fine-tuning process. To reduce computational cost, we leverage a scaled-down version with up to 1,000 instances per task, resulting in a total of 191,105 instances. MME (Fu et al., 2023a) is our comprehensive multimodal evaluation benchmark, measuring both multimodal perception and cognition capabilities across 14 subtasks. In addition, we further utilize 7 multimodal datasets for our evaluation. Specifically, for Optical Character Recognition, we utilize

Table S1: **Multimodal Dataset Details.**

| Dataset | Examples | Task Categories |
|---|---|---|
| Vision-Flan | 191K | Diverse |
| MME | 2374 | Diverse |
| Text-VQA | 5000 | OCR |
| VSR | 1222 | Spatial Reasoning |
| SNLI-VE | 17K | Visual Entailment |
| CIFAR-10 | 10K | Visual Perception |
| CIFAR-100 | 10K | Visual Perception |
| MNIST | 10K | Visual Perception |
| POPE | 9000 | Object Hallucination |

the *Text-VQA* (Singh et al., 2019), and for reasoning, we employ the Visual Spatial Reasoning (*VSR*) (Liu et al., 2023). Following (Zhai et al., 2023; Shen et al., 2024), the perception capability is tested on *CIFAR-10/100* (Krizhevsky et al., 2009) and *MNIST* (Deng, 2012). *SNLI-VE* (Xie et al., 2019) evaluates Visual Entailment capabilities, while the *POPE* (Li et al., 2023c) dataset examines the tendency towards object hallucination. The MME metric is the sum of accuracy values across all subtasks, while for the other 7 multimodal evaluation datasets, the metric used is just accuracy based on the assessment from Vicuna-13B-v1.5.

## S2 IMPLEMENTATION DETAILS

Following established practices in recent studies (Liu et al., 2024; Wang et al., 2024), we utilize the stage-one LLaVA (Liu et al., 2024) framework, incorporating CLIP-L (which consists of 24 Transformer-based encoder layers) as the vision encoder, along with a pre-trained cross-modality projector and Vicuna-7B-v1.3 (Chiang et al., 2023) (comprising 32 Transformer-based decoder layers) as the backbone LLM for our pre-trained LMM (refer to §3.1). The same editor architecture is implemented for both visual and multimodal representation editing (see §3.2). For visual representation editing, the entire visual representation in CLIP-L and the cross-modality projector layer is

modified. Notably, the visual representation from the second last vision encoder layer of CLIP-L is selected for fusion with textual representation in the stage-one LLaVA; thus, we omit the representation editor on the final vision encoder layer. In the case of multimodal representations, we apply edits to both textual-oriented prefixes and suffixes in Vicuna-7B-v1.3. For weights initialization, we initialize the low-rank matrix $U$ with orthogonal initialization, while the linear projector $Wx + b$ uses standard linear layer initialization in Pytorch (Paszke et al., 2019). For controllability experiment 4.3, we trained our two sets of representation editors $\psi_1$ and $\psi_2$ on the CIFAR-10 (Krizhevsky et al., 2009) training dataset for 1 epoch and evaluate the control performance on the testing dataset.

Table S2: **Hyperparameters and Configurations.**

| | |
|---|---|
| Learning Rate | $6e^{-4}$ |
| Batch Size | 128 |
| Epoch | 3 |
| Lr Scheduler | linear |
| Warmup Ratio | 0.03 |
| Activation Type | bfloat16 |
| Optimizer | Adam |

Additionally, during the fine-tuning, we focus on fine-tuning specific segments of the textual embeddings, particularly the prefix and suffix tokens, rather than the entire set of tokens. This decision is motivated by the role of these segments in Transformer-based decoder models. Prefix tokens are crucial for establishing the task-specific context early in the generation process, thereby conditioning the model's output effectively. Similarly, suffix tokens also play an important role in guiding and controlling generations due to the autoregressive training paradigm. To validate this design choice, we conducted an ablation study comparing different segment editing strategies in Table S3.

The results clearly demonstrate that fine-tuning both the prefix and suffix tokens yields the best performance, significantly outperforming the setting of fine-tuning all tokens. Specifically, we observe a substantial drop in the MME score when the entire textual embedding is edited (1233.90 $v.s.$ 1580.40). This suggests that over-editing the embeddings can lead to response drift, negatively impacting performance. This observation aligns with recent studies on prompt tuning (Han et al., 2024b; Lester et al., 2021; Oh et al., 2023; Mao et al., 2023), which indicate that larger adjustments (*i.e.*, longer inserted prompts in prompt tuning) do not necessarily lead to better performance and can, in fact, be less effective than smaller edits.

Table S3: **Edited Segments on Multimodal Representations.**

| Segments | MME |
|---|---|
| Prefix Only | 1465.32 |
| Suffix Only | 1497.35 |
| Prefix & Suffix | 1580.40 |
| All | 1233.90 |

## S3 EVALUATION METRICS

For a comprehensive evaluation, we utilize the MME benchmark (Fu et al., 2023b) alongside 7 additional multimodal datasets (see §S1). For MME, we employ the official evaluation tool (Yin et al., 2023), which includes both Perception and Cognition metrics. Specifically, MME covers existence, count, position, color, poster, celebrity, scene, landmark, artwork and OCR for perception and commonsense reasoning, numerical calculation, text translation and code reasoning for cognition. For the other 7 multimodal datasets, following (Shen et al., 2024), we use a consistent prompt template. This template incorporates the prompt, the model's prediction, and the ground truth for each test instance to guide Vicuna-13B-v1.5 (Zheng et al., 2024) in evaluating the accuracy of each prediction. We calculate the final accuracy on each multimodal dataset based on the percentage of Vicuna-13B-v1.5 judging "Yes."

Moreover, to further evaluate the effectiveness of MRT, we include two more multimodal benchmarks for comparison with two strong baselines on SEED (Li et al., 2023a) and GQA (Hudson & Manning, 2019) in Table S4, indicating that MRT consistently outperforms other PEFT approaches. We have also extended MRT to MiniGPT-v2 with EVA (Fang et al., 2023) as the vision encoder and LLaMA2-chat (7B) (Touvron et al., 2023) as the LLM, differing from the components of

Table S4: **More Zero-shot Evaluation.**

| Method | SEED-Bench | GQA |
|---|---|---|
| LLaVA$_{FT}$ | 57.4 | 53.5 |
| LoRA | 56.3 | 51.3 |
| MixLoRA | 55.9 | 52.2 |
| M$^2$PT | 57.1 | 50.3 |
| MRT | 57.6 | 52.7 |

Table S5: **Performance Comparison on MiniGPT-v2.**

| MiniGPT-v2 | MME |
|---|---|
| *FT* | 1464.88 |
| LoRA | 1358.14 |
| ReFT | 1346.65 |
| MixLoRA | 1418.48 |
| M$^2$PT | 1421.02 |
| MRT | 1439.73 |

LLaVA (Liu et al., 2024) in Table S5. Preliminary results on the MME benchmark demonstrate that MRT consistently achieves performance gains compared to other PEFT approaches.

## S4 MORE DIAGNOSTIC EXPERIMENT

To evaluate the significance of each component within MRT, we conduct comprehensive ablation experiments. In §4.4, we analyze the impact of removing each individual component from MRT. The results demonstrated that omitting any single component resulted in a noticeable performance drop (*e.g.*, a decrease to 1358 on MME when the multimodal editor was excluded), highlighting the importance of each part of the MRT framework. To further validate the effectiveness of MRT, we performed additional experiments by applying MRT to only a single component at a time (*i.e.*, LLM, Cross-modality, and Vision encoder). This approach allows us to understand the isolated impact of representation tuning on each modality component. As seen in Table S6, the best performance is achieved when MRT is applied to all components of the base Large Multimodal Model (LMM) simultaneously. This confirms that leveraging MRT across multiple components rather than focusing on a single modality leads to optimal improvements.

Table S6: **Impact of Components.**

| Component | MME | MMAvg |
|---|---|---|
| LLM | 1473.25 | 62.90 |
| Cross-modality | 1165.33 | 53.67 |
| Vision encoder | 1342.46 | 60.83 |
| All (MRT) | 1580.40 | 64.93 |

## S5 TRAINING AND INFERENCE TIME COMPARISON

As discussed in §2, although PEFT methods have generally been proven to be much more parameter-efficient compared to full fine-tuning in training, the burden of inference plays an important role in overall efficiency. Therefore, some studies (Lei et al., 2023; Han et al., 2024c) touch upon computational efficiency and potential impact on inference speed of PEFT methods. To further investigate the inference efficiency of our method, we conduct a comparison of PEFT methods in Table S7. Specifically, LoRA adds the minimum computational burden to inference with 12.5% incremental time, while MixLoRA introduces dynamic factor selection modules, which are more computational-intensive. Prompt-tuning (*i.e.*, M$^2$PT, VPT) employs extra prompts prepended with input sequences, costing significant inference overhead. It is worth highlighting that, our method represents a trade-off between inference time and performance, achieving significantly lower inference time increment (*e.g.*, 72.73% and 42.86% faster than the two most performance-competitive methods, M$^2$PT and MixLoRA) while reaching the highest performance on MME benchmark.

In addition, we include the memory usage and training time comparison in Table S8. It can be seen that MRT enjoys competitive training efficiency compared to existing PEFT approaches. We also want to highlight that both GPU memory usage and training time are lower than several baselines (*i.e.*, LoRA, MixLoRA).

Table S7: **Inference Time Comparison.**

| Method | MME | Inference Time | Increment |
|---|---|---|---|
| LLaVA$_{Align}$ | 1110.82 | 8 min | - |
| M$^2$PT | 1503.98 | 44 min | 450.0% |
| MixLoRA | 1509.61 | 21 min | 162.5% |
| VPT | 1398.74 | 17 min | 112.5% |
| LoRA | 1393.67 | 9 **min** | **12.5** % |
| MRT | **1580.40** | 12 min | 50.0% |

Table S8: **Training Efficiency Comparison.**

| Method | MME | # para | Memory Usage (GB) | Training Time (Hours) |
|---|---|---|---|---|
| LLaVA$_{Align}$ | 1110.82 | - | - | - |
| LLaVA$_{FT}$ | 1587.26 | 100% | 39 | 47 |
| VPT | 1398.74 | 0.06% | 12 | 7 |
| LoRA | 1393.67 | 0.63% | 19 | 16 |
| M$^2$PT | 1503.98 | 1.96% | 17 | 9 |
| MixLoRA | 1509.61 | 0.85% | 23 | 24 |
| MRT | **1580.40** | 0.03% | 16 | 9 |

## S6 EXTENDED CONTROLLABILITY EXPERIMENTS AND ANALYSIS

In this section, we provide further experimental analysis to evaluate the robustness and generalizability of our controllability framework. We present two key aspects: robustness of token-wise control and extension to the Text-VQA dataset. Additionally, we discuss potential directions for generalizing the framework using prompt engineering techniques.

### S6.1 ROBUSTNESS OF TOKEN-WISE CONTROL

Considering the textual question formats with the same semantic meaning can be vary. To achieve robust control, we introduce another multimodal representation editor by changing the textual prompt format from "Is the object an e in the image?" into "Is the object in the image an e?", trained under a similar setting as described in §4.3. Table S9 demonstrates that the new editors can achieve equally effective and robust control over counterfactual outputs. Additionally, to further enhance the generalizability of adapting various textual question formats, we leverage a lightweight rephraser to normalize different formats with the same semantic meaning into an expected template in §S6.3.

Table S9: **Controlled Counterfact Rate** with changed prompt format.

| Class $e$ (LLaVA$_{Align}$) | | Misclassification | | Misalignment | |
|---|---|---|---|---|---|
| | | Misclassfication on $e$ | *Others* | Misalignment to $\bar{e}$ | *Others* |
| (a) cat | 18.8% | 100% | 0% | 100% | 0% |
| (b) dog | 17.3% | 100% | 0% | 100% | 0% |
| (c) ship | 21.8% | 100% | 0% | 100% | 0% |
| (d) frog | 22.5% | 100% | 0% | 100% | 0% |
| (e) truck | 21.4% | 100% | 0% | 100% | 0% |

### S6.2 EXTENSION TO OTHER MULTIMODAL TASKS

We further extend MRT's controllability to tasks beyond image classification. We apply a similar strategy as outlined in §4.3 for Text-VQA (Antol et al., 2015). Specifically, we select 8,017 instances as the training set and 1,189 instances as the validation set on textual tokens beginning with "what is the $n$", where $n$ represents an image attribute (*e.g.*, name, color, brand). We aim to generate counterfactual outputs for Text-VQA. Different from the scenarios (*i.e.*, misclassification and misalignment §4.3) of

Table S10: **Controlled Counterfact Rate** on Text-VQA.

| Attribute ($n$) | Indeterminate |
|---|---|
| (a) name | 100% |
| (b) color | 100% |
| (c) brand | 100% |

counterfactual outputs on image classification task, we target the scenario of **Indeterminate** by altering the labels of all questions related to $n$ in the training set to *"Not sure"*. We train three distinct sets of representation editors $\psi = \{\psi_v^1, \psi_c, \psi_t^1\}$ for the attributes "name", "color", and "brand". Here, $\psi_v^1$ and $\psi_c$ are designed to edit only the image RoI (*i.e.*, the same setting in image classification), focusing on controlling key visual semantic information, while $\psi_t^1$ is trained to modify the token corresponding to $n$ (specifically, the 4th token in the sequence, as shown in the Text-VQA controllability pipeline Figure S1). The results in Table S10 indicate that our method successfully controls counterfactual outputs across various attributes. For instance, a question asking about the image "what is the name of this product?", the correct answer is "gum plus", our control leads the model to respond with *"Not sure"*, indicating indeterminacy of the attribute. In addition, Figure S2 shows some qualitative examples of counterfactual controls on Text-VQA.

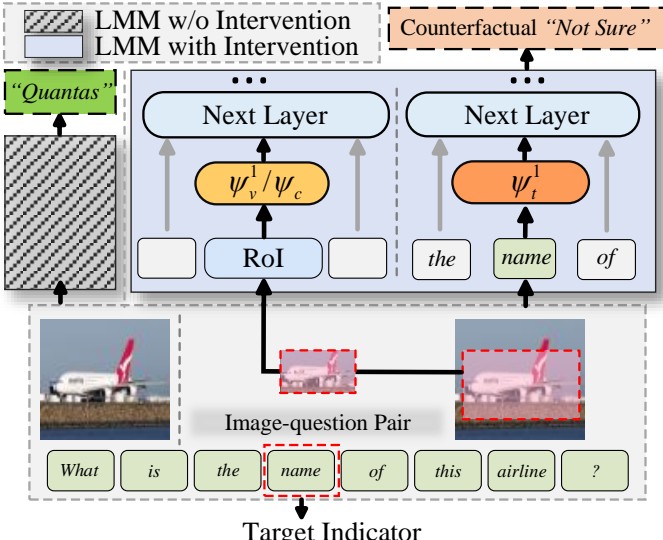

Figure S1: **Controllabilty Pipeline on Text-VQA.**

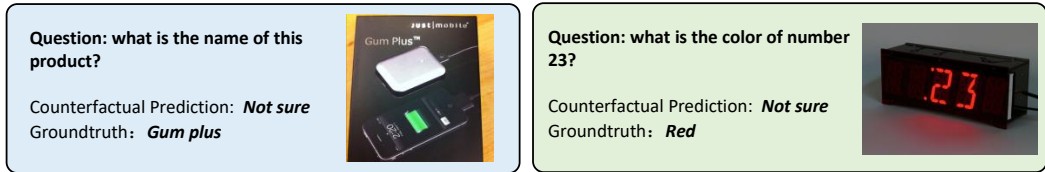

Figure S2: **Qualitative Examples** on Text-VQA.

### S6.3 GENERALIZABILITY DISCUSSION

Our current representation editors are effective in scenarios with fixed prompt formats. Considering the success of prompt engineering (White et al., 2023), crafting effective prompts to guide the output can further generalize the control across an even broader range of input queries, reducing sensitivity to variations in phrasing, and enhancing the robustness of MRT's controllability. For instance, different phrasings of a question (*e.g.*, "Is there an $e$ object visible in the image?" and "Does the image contain an object that is an $e$?") can be normalized into a standardized template (*e.g.*, "Is the object an $e$ in the image?"), making it possible for applying output control with our trained editors. Specifically, we leverage a lightweight rephraser based on T5-small (Raffel et al., 2020) (*i.e.*, 60M parameters), and customize a dataset for fine-tuning the rephraser, containing 6 different variant templates with the same semantic meaning of the expected input sequence. Table S11 shows that MRT can successfully achieve robust control on various input sequences with a single set of editors, even if they differ in lengths and structures.

Table S11: **Control Rate with Rephraser** on variant prompt formats.

| Prompt Formats | Output Control Rate |
| --- | --- |
| "Is there an $e$ object visible in the image?" | 100% |
| "Does the image contain an object that is an $e$?" | 100% |
| "Is the object shown in the image an $e$?" | 100% |
| "Is the object in the picture an $e$?" | 100% |
| "Do you recognize the object in the image as an $e$?" | 100% |
| "Can you tell if the object shown in the image is specifically an $e$?" | 100% |

## S7 ASSET LICENSE AND CONSENT

The majority of VPT (Jia et al., 2022) is licensed under CC-BY-NC 4.0. Portions of (Jia et al., 2022) are available under separate licenses: google-research/task_adaptation, huggingface/transformers, LLaVA and Vicuna are licensed under Apache-2.0; ViT-pytorch (Dosovitskiy et al., 2021) are licensed under MIT; LoRA is licensed under Contributor License Agreement (CLA). All the datasets included in our study are publicly available (*i.e.*, Vision-Flan, MME, Text-VQA, Visual Spatial Reasoning (VSR), CIFAR-10/100, MNIST, SNLI-VE, POPE), and all the models are publicly available. We would like to state that the contents in the dataset do NOT represent our views or opinions.

## S8 REPRODUCIBILITY

MRT is implemented in Pytorch (Paszke et al., 2019). Experiments are conducted on NVIDIA A100-40GB GPUs. Our full implementation is available at `https://github.com/comeandcode/MRT`.

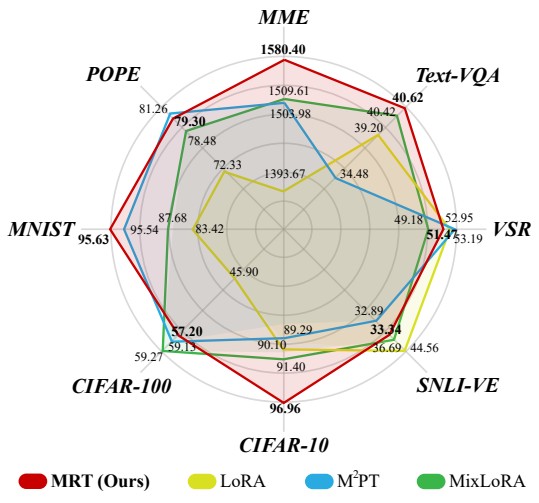

Figure S3: **Performance Comparison.**

## S9 SOCIAL IMPACT AND LIMITATIONS

This study presents MRT, demonstrating significant performance enhancements (see Figure S3) with low parameter usage and fundamental insights into LMM controllability. Our approach is particularly valuable in real-world, computation-sensitive applications, *e.g.*, training machine learning models on edge devices. MRT investigates LMM controllability from a casual model perspective (Geiger et al., 2021). One step further, if the MRT's causal structure can be explicitly defined, we may pave the way towards *ad-hoc* interpretability, which is crucial for the continuous development of PEFT across a wider spectrum of trustworthy applications.

For potential limitations, our method brings a hyperparameter — rank (*i.e.*, low-rank matrix $U$ in Eq. 1), which directly determines the number of tunable parameters. Similar to other low-rank approaches (Hu et al., 2022; Shen et al., 2024), it notably correlated to the MRT's performance (see

§4.4). Though during the experiment, we found that the optimal results fall into a relatively small range (*i.e.*, rank 2-8 for both visual-based and multimodal editors), current manual searching on ranks might be time insufficient. Introducing a small network within the MRT to autonomously search for optimal combinations might enhance training efficiency and facilitate additional performance improvements (Han et al., 2023).

## S10 ETHICS CONCERNS

The inherent design of MRT, characterized by the utilization of semantic representations, alongside with the token-wise controllablity, implies its capability of manipulating the model generation. Our approach offers promising avenues for enhancing large multimodal models (LMMs). However, the real-world application of such models necessitates careful consideration of ethical implications, including the potential for misinformation, privacy violations, harmful content generation, and the amplification of biases. Therefore, the appropriate employment of MRT is crucial to equip LMMs with the ability to generate reliable, controllable, and high-quality content.

Additionally, there are possible misuse scenarios and corresponding mitigation strategies. First, attackers can manipulate models to produce misinformation (*e.g.*, misclassification) via intentionally altering the model's understanding of an input image (Chen et al., 2021). Second, biased information can be produced or amplified. Attackers can edit the textual tokens related to sensitive attributes in the multimodal representation, leading to harmful or discriminatory outputs (D'Incà et al., 2024). In order to mitigate possible misinformation, we suggest performing adversarial robustness testings (Dong et al., 2022) that explicitly check for consistency in object recognition across varying queries. For mitigating bias generation or amplification, one solution can be bias detection and correction procedure on generated content, monitoring the representation for bias patterns and applying corrective measures if detected (D'Incà et al., 2024). Another solution lies in clearly documenting any controlled editing made to the model's representation and disclosing any potential biases introduced during this process (Shah & Sureja, 2024). In conclusion, while MRT exhibits strong output controllability, applying MRT to realistic applications still requires ethical safeguarding, robust testing, and transparency measuring. From a security perspective, MRT presents significant potential, as it may facilitate the development of white-box attack and defense strategies tailored to LMMs.

## S11 DISCUSSION AND FUTURE WORK

While representation tuning has been explored in the NLP field (Wu et al., 2024a;b), we would like to highlight three key technical contributions of MRT specifically tailored to the multimodal domain.

**First, intuitive yet effective control.** MRT is the first attempt to enable token-wise control over LMMs through representation editing. By directly editing the semantic information of the image RoI and the textual target class indicator token, MRT offers an interpretable and intuitive mechanism for adjusting model predictions. This level of fine-grained controllability is difficult to achieve with existing baselines.

**Second, loss optimization.** From an optimization perspective, we provide a detailed analysis of why MRT outperforms other PEFT methods. By visualizing the loss landscape, we demonstrate that multimodal representation tuning enhances the generalization capabilities of LMMs, highlighting a promising direction for future PEFT research.

**Third, joint multimodal learning.** Unlike single-modality research, multimodal settings require consideration of two additional factors: **multimodal integration** and **vision modality editing**. To address this, we designed a framework that optimizes the cross-modality layer to effectively bridge the gap between the two modalities. While current PEFT approaches (Shen et al., 2024; Wang et al., 2024; Hu et al., 2022; Han et al., 2024b) for LMMs typically unfreeze the cross-modality projector during stage-2 tuning, we adhere to the principle of representation editing by introducing a lightweight cross-modality editor, achieving significantly lower parameter usage while delivering substantial performance gains. For vision modality editing, MRT takes a markedly different approach from current NLP practices by focusing on editing all visual representations. This method highlights the sparsity of visual information and suggests that broader editing strategies should be explored in the vision domain.

Despite MRT's systemic efficiency and effectiveness, it also comes with new challenges and unveils some intriguing questions. For example, as mentioned in Appendix §S9, the ranking for MRT is currently governed by manually defined values (see §4.4), although we do not need to specify prompt lengths as required by *prompt tuning* methods (*e.g.*, M$^2$PT, VPT). Another essential future direction deserving of further investigation is the LMM controllability. In §3.3 and §4.3, we demonstrate that effectively intervening in only a few targeted instrumental visual-based and multimodal tokens can generate semantically counterfactual outputs. This intriguing observation is inherently linked to network attacks (Li et al., 2022; Guo et al., 2022; Saha et al., 2020), as one can readily compromise the model's performance, indicating that the multimodal framework may be susceptible to disruption. The applicability of this direction needs further investigation. Moreover, although we have conducted the optimization analysis based on loss landscape (Li et al., 2018a; Ma et al., 2022), currently the community does not have a standard evaluation metric that we can follow. Therefore, we plan to conduct further theoretical analysis, including how the incorporation of representation editing influences the attention module (*e.g.*, attention activation pattern analysis (Wang et al., 2024)) and gradient flow analysis (Bambhaniya et al., 2024).

