# OpenReview forum: "Re-Imagining Multimodal Instruction Tuning: A Representation View"
_ICLR.cc/2025/Conference — ICLR 2025 Poster_

### Official Review · Reviewer_RZ9b · 2024-10-30

**Soundness:** 2
**Presentation:** 2
**Contribution:** 2
**Rating:** 5
**Confidence:** 5

**Summary:**

This paper adopted the parameter-efficient fine-tuning method Representation Tuning to the multimodal large language model domain. This paper used different representation editors for the vision encoder, LLM, and cross-modality projectors to optimize the visual representation, cross-modality representation, and multimodal representation. Experiments on several MLLM and image classification benchmarks show the efficiency and effectiveness of the proposed method. The paper further conducted controllability studies on image classification benchmarks to show the possible controllability of the proposed methods. Extensive ablation studies are furhter discussed for the designs of several hyper-parameters of the proposed method.

**Strengths:**

1. This paper provides a promising and efficient PEFT method for MLLMs as an alternative to the commonly used LoRA-based methods.
2. The ablation studies are comprehensive for a better understanding of the proposed method and hyper-parameter design choices.

**Weaknesses:**

1. The paper directly adapts the representation learning method in LLM to the MLLM in a rather straightforward way thus the technical contribution is limited.
2. The benchmark selection for comparisons is not comprehensive and convincing enough. MME is a relatively small-scale MLLM benchmark that has non-trivial variances. The paper should include more comprehensive and commonly used multimodal benchmarks like SEED, MMBench, MMMU, GQA, VQAv2, ChartQA, and DocVQA for more convincing comparisons.
3. The performance of other methods might have some problems. In Table 1 of this paper, the lora baseline is significantly worse than the full-finetuning one, but according to LLaVA's results (https://github.com/haotian-liu/LLaVA/blob/main/docs/MODEL_ZOO.md), the performances should be similar. Besides, the proposed method adds tunable parameters in the vision encoder while other baselines use frozen vision encoders. Baselines with unfrozen vision encoders should also be added for comparison.
4. As the proposed method brings additional parameters and time costs in the inference phase, the efficiency advantage of the proposed method should be clearly verified in the training phase, including GPU memory usage and training speed comparisons,
5. To validate the generalization ability of the proposed method, the authors should include experiments with different vision encoders + different LLM types and sizes.
6. Although the paper claimed that the proposed method brings more interpretability and controllability to MLLMs compared with common practices, it is hard to see how could the method really help the interpretability and controllability of MLLMs. The controllability study is too customized with human priors and provides little help for MLLM controllability in the general case.

**Questions:**

1. In Sec 3.2 Cross-modality Representation part, it says that the projector integrates representations from each layer of the visual encoder. Does this mean you combine vision features in different layers in the vision encoder as the final vision input for LLM?
2. What is the reason for applying prefix and suffix editors on textual tokens, as the prefix and suffix of different textual prompts at the same position might have very different semantics and meanings? Wouldn't this be harmful for the claimed interpretability and controllability?

---

> ### Author Response · Authors · 2024-11-20
> **To reviewer RZ9b (Part I)**
>
> Dear Reviewer RZ9b,
>
> We sincerely thank you for the valuable time and constructive feedback, which are crucial for improving our work. We provide explanations to each question as follows.
>
> **Q1: Regarding MRT’s technical contributions.**
>
> **A1:** While representation tuning has been explored in the NLP field, we would like to highlight three key technical contributions of MRT specifically tailored to the multimodal domain.
>
> **First, intuitive yet effective control.** MRT is the first attempt to enable token-wise control over LMMs through representation editing. By directly editing the semantic information of the image RoI and the textual target class indicator token, MRT offers an interpretable and intuitive mechanism for adjusting model predictions. This level of fine-grained controllability is difficult to achieve with existing baselines.
>
> **Second, representation learning loss optimization.** From an optimization perspective, we provide a detailed analysis of why MRT outperforms other PEFT methods. By visualizing the loss landscape, we demonstrate that multimodal representation tuning enhances the generalization capabilities of LMMs, highlighting a promising direction for future PEFT research.
>
> **Third, joint multimodal learning.** Unlike single-modality research, multimodal settings require consideration of two additional factors: **multimodal integration** and **vision modality editing**. To address this, we designed a framework that optimizes the cross-modality layer to effectively bridge the gap between the two modalities. While current PEFT approaches [ref1, ref13] for LMMs typically unfreeze the cross-modality projector during stage-2 tuning, we adhere to the principle of representation editing by introducing a lightweight cross-modality editor, achieving significantly lower parameter usage while delivering substantial performance gains. For vision modality editing, MRT takes a markedly different approach from current NLP practices by focusing on editing all visual representations. This method highlights the sparsity of visual information and suggests that broader editing strategies should be explored in the vision domain.
>
> Thank you again for the great question. We have supplemented the above discussions in Appendix S11.
>
>
> **Q2: Regarding the scale of the benchmarks.**
>
> **A2:** Thank you for the excellent suggestion. In our work, we evaluated MRT on seven additional larger scale datasets beyond MME, encompassing a total of 63,123 instances. To further strengthen our evaluation, we followed your suggestion and conducted additional experiments on the SEED-Bench and GQA benchmarks. As shown in the results below, MRT consistently outperforms other PEFT approaches. We’ve included the detailed discussions and the corresponding results in the revised paper (see Appendix S3).
>
> | Method  | SEED-Bench | GQA |
> | ---- | ----------- | --------------- |
> | MixLoRA  |   55.9     |      52.2      |
> | M$^2$PT |    57.1    |     50.3       |
> | MRT         |  57.6      |      52.7      |
>
>
> **Q3.1: Regarding the LoRA performance.**
>
> **A3.1:** Thank you for the insightful observation. The primary reason for the performance difference lies in the training data used. Specifically, our work utilizes the Vision-Flan dataset [ref11], following previous studies [ref1-2]. In contrast, the LoRA results from LLaVA were obtained using LLaVA’s training data (https://huggingface.co/liuhaotian/llava-v1.5-7b-lora). However, our reported LoRA results align with those in MixLoRA [ref2] and M$^2$PT [ref1], which are also trained on the Vision-Flan dataset. We’ll make this more clear in the revised paper.
>
> **Q3.2: Baselines with unfrozen vision encoders should also be added for comparison.**
>
> **A3.2:** Thank you for the excellent suggestion. We would like to clarify that MRT has been compared with M$^2$PT [ref1] and VPT [ref8] in Table 1, both of which are PEFT methods utilizing tunable parameters in the vision encoder. Specifically, M$^2$PT introduces tunable soft prompts into both the frozen vision encoder and the frozen backbone LLM, while VPT focuses solely on inserting trainable prompts into the frozen vision encoder. Experimental results show that MRT delivers noticeably superior performance compared to M$^2$PT and VPT.

---

> ### Author Response · Authors · 2024-11-20
> **To reviewer RZ9b (Part II)**
>
> **Q4: Memory & Time Efficiency in training.**
>
> **A4:** Following the suggestion, we have included the efficiency in the training stage w.r.t. trainable parameters, memory usage, and training time in the table below. It can be seen that MRT enjoys a competitive training efficiency with existing PEFT approaches. We also want to highlight that both GPU memory usage and training time are lower than several baselines (i.e., LoRA, MixLoRA). For completeness, we have added these results to Appendix S5 in the revised paper. Thank you again for the great suggestion.
>
> | Method  | trainable params  | Memory Usage (GB)* | Training Time (Hours) | MME  |
> | ---- | ---- | ----------- | --------------- |--------------- |
> | VPT  | 0.06%  |   12   |     7       |     1398.74       |
> | MixLoRA  | 0.85%  |  23     |      24      |   1509.61         |
> | M$^2$PT | 0.09%  |   17     |      9      |    1503.98        |
> | MRT         |  0.03%  |  16    |     9      |     1580.40       |
>
> *Peak under the same batch size setting.
>
>
> **Q5: Apply MRT to different LMMs.**
>
> **A5:** We completely agree that experimenting with different LMMs can enhance the generalizability of our method. Following the suggestion, we conducted additional experiments using a different LMM configuration, MiniGPT-v2 [ref3] with EVA [ref4] as the vision encoder, a linear projection layer as the cross-modality module, and LLaMA2-chat (7B) [ref5] as the LLM, differing from the components of LLaVA. For these experiments, we selected stage-2 MiniGPT-v2 without multimodal instruction tuning as the backbone model and used a scaled-down version of the Vision-Flan dataset, consisting of 191K instances, for fine-tuning. Preliminary results on the MME benchmark demonstrate that MRT consistently achieves performance gains compared to other PEFT approaches. We will include more comprehensive results and discussions in the revised paper.
>
>
> | MiniGPT-v2 | MME |
> | ---- | ----------- |
> | ReFT  |   1346.65     |
> | MixLoRA  |   1418.48     |
> | M$^2$PT |    1421.02     |
> | MRT         |   1439.73     |
>
>
> **Q6: Interpretability and controllability.**
>
> **A6:** Thank you for raising this question. We would like to provide some clarification on how the proposed method helps both interpretability and controllability in MLLMs and how it can generalize beyond the specific scenarios studied.
>
> For interpretability, the proposed method introduces token-wise editing, which allows for direct and targeted manipulation of specific semantic representations, such as the image RoI and the textual target indicator token. By focusing on these tokens, the impact of edits becomes transparent and traceable.
>
> For controllability, we have conducted more controllability experiments on **robustness of token-level control**, **extending to other multimodal tasks**, **generalizability**, as well as including further discussion of employing prompt engineering to allow control across an even broader range of input queries. Please kindly refer to Appendix S6 for the full details. Thank you!
>
> **Q7: Regarding combining vision features in different layers.**
>
> **A7:** Sorry for the confusion. We want to clarify that MRT does not combine vision features in different layers in the vision encoder as the final input for LLM. In Appendix S2, we have included the discussion that only the visual representation from the second last encoder layer is selected and fed into the cross-modality projector layer in order to be concatenated with textual tokens. This step strictly follows the common setting of stage-one LLaVA [ref1-2]. We have also revised line 218-219 to prevent any confusions.

---

> ### Author Response · Authors · 2024-11-20
> **To reviewer RZ9b (Part III)**
>
> **Q8.1: The reason for applying prefix and suffix editors on textual tokens.**
>
> **A8.1:** We would like to explain the reason for choosing prefix and suffix tokens. Prefix tokens are vital for conditioning the model to specific tasks or behaviors [ref10, ref12], while suffix tokens are significant in shaping and directing output due to the autoregressive nature of the models. Acknowledging their importance in the training process, we prioritize editing these tokens within the textual embeddings.
>
> Our additional experiments on editing various segments of the text-based representations show that fine-tuning both prefix and suffix tokens leads to the optimal performance. Specifically, the decrease (i.e., 1233.90 compared to 1580.40) on the MME score when editing all textual tokens indicates that excessive editing of the embeddings can cause response drift, adversely affecting performance. This observation is also consistent with recent research on prompt tuning [ref1, ref6-7], which suggests that larger modifications (i.e., longer soft prompts in prompt tuning) do not necessarily improve performance and may even be less effective than smaller ones.
>
> | Segments  | MME |
> | ---- | ----------- |
> | Prefix Only  |   1465.32     |
> | Suffix Only |    1497.35    |
> | Prefix & Suffix    |  1580.40      |
> | All         |  1233.90      |
>
>
> **Q8.2: Applying prefix and suffix editors on textual tokens may be harmful for controllability.**
>
> **A8.2:** We want to clarify that the editing within textual-oriented representations during training and controlling are taken place in separate positions, which does not have a harmful impact for the claim. Specifically, for training, we only edit prefix and suffix locations, which is motivated by their critical roles in establishing general task context and guiding generation. Previous works have shown that editing these locations delivers the most significant gains [ref9]. For controlling, instead of intervening semantic ambiguous positions, we precisely focus on the targeted textual-oriented token as it contains direct and interpretable impact to the textual questions.
>
> To further address your concern on the generalization of the editing, during the rebuttal phase, we further change the order of text instruction for different template in (Appendix S6), and other counterfactual controls (i.e., on Text-VQA, also see Appendix S6). These results consistently demonstrate the robustness and effectiveness of our controlling strategy on textual inputs.
>
> [ref1] Taowen Wang, et al. M$^2$PT: Multimodal prompt tuning for zero-shot instruction learning. EMNLP, 2024.
>
> [ref2] Ying Shen, et al. Multimodal instruction tuning with conditional mixture of lora. ACL, 2024.
>
> [ref3] Chen, Jun, et al. MiniGPT-v2: large language model as a unified interface for vision-language multi-task learning, ArXiv, 2023.
>
> [ref4] Yuxin Fang, et al. Eva: Exploring the limits of masked visual representation learning at scale. ArXiv, 2022.
>
> [ref5] Hugo Touvron, et al. Llama 2: Open foundation and fine-tuned chat models. ArXiv, 2023.
>
> [ref6] Changdae Oh, et al. Blackvip: Black-box visual prompting for robust transfer learning. ArXiv, 2023.
>
> [ref7] John Merrill, et al. Doubly right object recognition: A why prompt for visual rationales. ArXiv, 2022.
>
> [ref8] Cheng Han, et al. Facing the elephant in the room: Visual prompt tuning or full finetuning? ICLR, 2024.
>
> [ref9] Zhengxuan Wu, et al. Reft: Representation Finetuning for Language Models. NeurIPS, 2024
>
> [ref10] Raffel, Colin, et al. Exploring the limits of transfer learning with a unified text-to-text transformer. Journal of machine learning research 21.140 (2020): 1-67.
>
> [ref11] Xu, Zhiyang, et al. Vision-flan: Scaling human-labeled tasks in visual instruction tuning. ACL, 2024.
>
> [ref12] Bavarian, Mohammad, et al. Efficient training of language models to fill in the middle. ArXiv, 2022.
>
> [ref13] Zhou, Xiongtao, et al. An Empirical Study on Parameter-Efficient Fine-Tuning for MultiModal Large Language Models. 2024.
>
> We sincerely appreciate your thoughtful comments. We hope our response addresses your concerns. Please let us know if there are any additional questions, and we will be happy to discuss further.

---

> ### Author Response · Authors · 2024-11-25
> **Looking forward to the discussion**
>
> Dear Reviewer RZ9b,
>
> We deeply appreciate the time and effort you’ve taken to review our work, especially given your busy schedule. As the authors-reviewer discussion phase draws to a close, we would be grateful for the opportunity to engage in dialogue with you. Our goal is to ensure we've adequately addressed your concerns and welcome any additional questions or points of discussion you'd like to raise.
>
> Thank you for your thoughtful consideration.
>
> Best regards,\
> The Authors

---

> > ### Comment · Reviewer_RZ9b · 2024-11-26
> > **Response to rebuttal**
> >
> > Thanks for the authors' detailed responses. Part of my concerns are addressed. I still have concerns about the generalizability of the proposed method as the authors suggest in Q3.1 that different training data would lead to significant performance differences for some methods like the vanilla Lora. Besides, I appreciate that the authors have provided experiment results on additional benchmarks, but it would be better to provide the full-finetuning and Lora-tuning results for reference as well. Therefore, I will only increase my score to 5.

---

> ### Author Response · Authors · 2024-11-26
> **Thank you for the prompt response**
>
> Dear Reviewer RZ9b,
>
> We sincerely appreciate your engagement in the discussion and your valuable feedback, which are crucial in enhancing the quality of our work. We are pleased that our response addresses most of your concerns and would like to take this opportunity to provide additional results based on your suggestions.
>
> **Regarding the performance of vanilla Lora.**
>
> Thanks for the additional question. We’d like to provide some explanations here.
>
> **First**, we completely agree that the performance gap between full fine-tuning and LoRA is small with the original LLaVA’s dataset. In fact, as shown in our results, LoRA **also achieves comparable or even superior performance** to full fine-tuning on the Text-VQA, VSR, and SNLI-VE datasets. However, the performance gap is more pronounced on benchmarks such as MME, CIFAR-100, MNIST, and POPE. We hypothesize that this discrepancy arises because the Vision-Flan dataset (191K) is relatively smaller and differs in data quality and distribution compared to LLaVA’s dataset (665K). Similar performance gaps between full fine-tuning and LoRA have been observed in MixLoRA (see [Table 1](https://aclanthology.org/2024.acl-long.38.pdf)) and M$^2$PT (see [Table 1](https://aclanthology.org/2024.emnlp-main.218.pdf)), both of which were also trained on Vision-Flan.
>
> **Second**, to validate this further, we conducted an additional experiment using LLaVA’s dataset. The results on the MME benchmark confirm that the performance gap between full fine-tuning and LoRA is effectively closed when trained on this dataset. Meanwhile, MRT consistently outperforms LoRA, maintaining its superior performance.
>
> | LLaVA-v1.5 7B | MME |
> | ---- | ----------- |
> | Full Fine-tuning  |   1510.7     |
> | LoRA  |   1476.9     |
> | MRT   |    1486.7     |
>
> **Regarding full-finetuing and lora results on the additional experiments.**
>
> Following your suggestion, we have included the results of full fine-tuning and LoRA-tuning on additional benchmarks for completeness. The findings show that there is no large performance gap between LoRA-tuning and full fine-tuning on the SEED and GQA benchmarks, while MRT consistently delivers the best performance. Similarly observations are seen on MiniGPT-v2.
>
> | Method  | SEED-Bench | GQA |
> | ---- | ----------- | --------------- |
> | **Full Fine-tuning**  |  57.4     |    53.5       |
> | **LoRA**                  |   56.3    |     51.3      |
> | MixLoRA            |   55.9     |      52.2      |
> | M$^2$PT            |    57.1    |     50.3       |
> | MRT               |    57.6    |      52.7      |
>
> | MiniGPT-v2 + EVA | MME |
> | ---- | ----------- |
> | **Full Fine-tuning**  |   1464.88    |
> | **LoRA**  |    1358.14   |
> | ReFT  |   1346.65     |
> | MixLoRA  |   1418.48     |
> | M$^2$PT |    1421.02     |
> | MRT         |   1439.73     |
>
> We further include the training efficiency analysis for full fine-tuning and LoRA-tuning. MRT demonstrates competitive training efficiency: It offers lower GPU memory usage and reduces training time compared to several baselines, such as LoRA and MixLoRA.
>
> | Method  | trainable params  | Memory Usage (GB)* | Training Time (Hours) | MME  |
> | ---- | ---- | ----------- | --------------- |--------------- |
> | **Full Fine-tuning** | 100%  |  39      |   47       |    1587.26     |
> | **LoRA**  | 0.63%  |   19   |     16       |     1393.67       |
> | VPT  | 0.06%  |   12   |     7       |     1398.74       |
> | MixLoRA  | 0.85%  |  23     |      24      |   1509.61         |
> | M$^2$PT | 0.09%  |   17     |      9      |    1503.98        |
> | MRT         |  0.03%  |  16    |     9      |     1580.40       |
>
> We greatly value each comment and suggestion from your review, and are hoping that our additional clarifications and experimental results address your concerns. We are eager to address any remaining issues during the discussion phase. Thank you very much.
>
> Best Regards,\
> Authors

---

> ### Author Response · Authors · 2024-12-02
>
> Dear Reviewer RZ9b,
>
> We've updated the revised paper based on your suggestions by adding the full fine-tuning and LoRA results to those additional experiments.
>
> As the end of the discussion phase is approaching, we would be truly grateful if you could inform us whether our recent response has adequately addressed your additional questions. Your feedback is invaluable to us and plays a critical role in enhancing the quality of our work. We deeply appreciate the effort and time you have dedicated to reviewing our paper.
>
> Best Regards,\
> Authors

---

### Official Review · Reviewer_PDvR · 2024-11-02

**Soundness:** 3
**Presentation:** 3
**Contribution:** 3
**Rating:** 6
**Confidence:** 3

**Summary:**

This paper introduces a method for tuning large multi-modal models (LMM) in a efficient but effective way so that it can achieve similar performance to full fine-tuning, with an additional objective of having a controllability. The key idea of this paper is based on a prior technique that learns parameters that edits the representations. The main contribution of this paper is to make use of this technique for efficient tuning of LMMs, and the experiments show that this idea indeed is helpful and the paper did a good job of investigating the effect of various design choices.

**Strengths:**

- Motivation is clear
- Intuitive and simple idea that works well
- Extensive analysis/ablation on the design choices

**Weaknesses:**

Experiments on controllability is interesting but not conclusive. For instance,
- What would happen if you don't use ROI and do train with all tokens?
- What would happen if you fine-tune other baselines for this setup?
- What would happen if you use a sentence that has a same semantic meaning to 'Is the object an e in the image?' but with different structure and words, after fine-tuning? Would the model still be controlled as intended?

Additional weaknesses are:
- Figure 5 on the optimization landscape is interesting but I'm not sure how it is cherry-picked. Would there be a way to make this claim be supported by some metrics or more figures?
- Main method section feels a bit redundant to me, not much of a difference between each subsection. It could be nice to think of a better way to re-structure, remove redundancy, and think of a way to clearly explain what differences exists

Note: I'm not an expert in this area so I might be missing some experimental details. I'll check the other reviews on how the experimental setup is valid and how the results are good compared to other baselines.

**Questions:**

See Weaknesses

---

> ### Author Response · Authors · 2024-11-19
> **To Reviewer PDvR (Part I)**
>
> Dear Reviewer PDvR,
>
> We sincerely thank reviewer PDvR for the valuable time and constructive feedback! We provide explanations to your questions point-by-point in the following.
>
> **Q1: Don’t use RoI but train with all tokens.**
>
> **A1:** Although editing more tokens (i.e., all visual tokens) does not prevent the model from producing expected counterfactual results, it becomes difficult to isolate the impact of the edits on specific semantic features of the target object, leading to a severe loss of interpretability. In contrast, targeting the RoI, combined with edits to the textual target indicator token (i.e., class token e), ensures that the edits are directly tied to the most relevant semantic features of the object and the query, preserving interpretability. Moreover, from the training efficiency perspective, targeting the RoI reduces the computational complexity and speeds up convergence.
>
>
> **Q2: Controllability results of other baselines.**
>
> **A2:** With the same setups, although other baselines could also learn the pattern and produce counterfactual results, they are not able to achieve token-wise controllability. This limitation arises because other baselines typically operate in a black-box manner, where the manipulations are based on implicit adjustments by injecting new prompts (i.e., prompt tuning) or by modifying the internal weights through low-rank updates (i.e., LoRA) without providing explicit control over representations. On the contrary, our approach achieves token-wise controllability by introducing dedicated representation editors that are trained to selectively edit the most semantically relevant tokens (i.e., visual RoI and textual class token). This fine-grained editing mechanism ensures that the impact of the edits is directly tied to specific features, enabling interpretable control over the model’s counterfactual outputs.
>
>
> **Q3: Different structure and words for intended control.**
>
> **A3:** This is a great question! Prompts with different structures or phrasings will not break the controllability of MRT. MRT is capable of handling variations in prompt formats by training control editors tailored to the new prompt structures or phrasings. To validate this capability, we conducted a controllability experiment where the textual prompt “Is the object an e in the image?” is altered to “Is the object in the image an e?” and trained control editors using the same setup as described in Sec. 4.3. The results, shown in the table below, confirm that the newly trained editors can achieve equally robust and effective control over counterfactual outputs.
>
> | Class e | Misclassification rate on e | Misalignment rate on e |
> | ---- | ----------- | --------------- |
> | dog  |    100%        |     100%       |
> | cat |    100%      |     100%       |
> | Ship |   100%     |     100%       |
> | Frog |    100%    |    100%        |
> | Ship |   100%     |    100%        |
>
> To further enhance the robustness of MRT’s controllability when considering diverse or complex textual instructions, we plan to incorporate prompt engineering [ref1], which can normalize different phrasings with the same semantic meaning into standardized templates. For example, sentences such as “Can you identify the animal in the image?” and “Do you know what kind of animal is depicted here?” could be normalized to a standard template as “What is the animal in the image?” This approach would allow us to apply the trained editors consistently across a broader range of inputs, enhancing the robustness and generalizability of the controllability framework.
>
> We have added additional discussions to the revised paper (see Appendix S6). Thank you!

---

> ### Author Response · Authors · 2024-11-19
> **To Reviewer PDvR (Part II)**
>
> **Q4: Optimization landscape.**
>
> **A4:** Thank you for your positive assessment. We want to address your concerns from two perspectives.
>
> **How do we pick the optimization landscape?**
> It is not a cherry-picked landscape, but instead a general visualization. Figure 5 illustrates the 2D/3D loss surfaces [ref2] for three best models based on the evaluation results. Specifically, we randomly pick up two directions of changing the weights and randomly select a subset of training data for plotting loss surfaces for all three models. This randomness, as stated in various works [ref2-4], does not significantly affect the results. MRT provides a flatter loss surface, indicating that MRT is less sensitive to loss fluctuations, leading to better generality compared to other approaches’ loss landscapes.
>
> **Quantitative or qualitative evaluation.**
> We agree that a systemic analysis quantitatively or qualitatively can further strengthen our claim. Unfortunately, currently the community does not have a standard evaluation metric that we can follow. However, we try to explain the loss landscape further, and highlight a promising direction of studying the loss landscape. [ref2] claims that a sharp loss surface near the minimum indicates that small perturbations in the weights can lead to a significant increase in loss. This suggests the model may generalize poorly, as it can be understood as a sign of overfitting to the training data. A flatter loss surface around the minimum, on the other hand, typically suggests better generalization. This is because the model is more robust to small changes in the weights, indicating a more stable solution has been learned.
>
> We have added more discussions w.r.t. optimization landscape in the revised paper. Thank you!
>
> **Q5: Main method is a bit redundant.**
>
> **A5:** Thank you for your suggestion. We have revised the paper accordingly.
>
> [ref1] White, J., et al. A prompt pattern catalog to enhance prompt engineering with chatgpt. ArXiv, 2023.
>
> [ref2] Hao Li, et al. Visualizing the loss landscape of neural nets. NeurIPS, 2018.
>
> [ref3] Runjia Zeng, et al. Visual Fourier Prompt Tuning. NeurIPS, 2024.
>
> [ref4] Ma, Xu, et al. Rethinking network design and local geometry in point cloud: A simple residual MLP framework. ICLR, 2022.
>
>
>
> We sincerely appreciate your thoughtful comments. We hope our response addresses your concerns. Please let us know if there are any additional questions, and we will be happy to discuss further.

---

> > ### Comment · Reviewer_PDvR · 2024-11-25
> >
> > Thank you for your response. I read the response other review. I agree with Reviewer RZ9b in that the controllability experimental setup is a bit contrived, and it could be nice to think of how to design more natural setups. But I still think this paper deserves the score of 6 so I maintain my score.

---

> > > ### Author Response · Authors · 2024-11-25
> > > **Thank you for your response**
> > >
> > > We sincerely thank the reviewer for their prompt response and thoughtful feedback. To address controllability, we have included additional experiments in Appendix S6, covering the _robustness of token-level control_, _extensions to other multimodal tasks_, and _generalizability_. Additionally, we have provided further discussion on employing prompt engineering to enable control across a broader range of input queries. We are excited to explore this direction further in our future work. Thank you once again for the valuable suggestion and for your positive assessment of our work.

---

### Official Review · Reviewer_Yzen · 2024-11-04

**Soundness:** 3
**Presentation:** 3
**Contribution:** 3
**Rating:** 6
**Confidence:** 3

**Summary:**

This paper introduces Multimodal Representation Tuning (MRT), a parameter-efficient fine-tuning method to enhance controllability and interpretability in multimodal large language models (LMMs). MRT addresses the challenge of adapting LMMs effectively with fewer parameters by leveraging token-level multimodal representation control, achieving superior performance with up to 21 times fewer parameters than similar approaches. The authors explore and improve model behavior control through MRT, illustrating benefits in various multimodal perception and cognition benchmarks.

**Strengths:**

- MRT is intuitive, simple, and effective. It uses significantly fewer parameters while achieving strong results, making it suitable for resource-constrained applications.
- The approach provides granular control over the representation editing, enabling counterfactual output generation that enhances interpretability.
- The method is validated across multiple multimodal tasks, illustrating its effectiveness in diverse domains such as OCR, visual perception, and spatial reasoning.
- The paper is well-written and easy to follow.

**Weaknesses:**

- The rank parameter is integral to MRT’s performance, but it currently requires manual tuning, which may limit practical adoption. While promising, the need for automated rank selection is highlighted as a limitation, suggesting a more autonomous rank-searching mechanism that could enhance usability.
- The empirical performance is decent, but could you elaborate on MRT's main contribution compared to ReFT instead of extending the interchange intervention idea into MLLM?
- Given its control over multimodal representations, MRT’s potential for misuse (e.g., manipulation of outputs) is acknowledged but not fully addressed in terms of mitigation strategies. Could you provide some examples of possible misuse of MRT utilizing controllability?

**Questions:**

1. Could the authors elaborate on any plans to automate the rank-tuning process within MRT to simplify its application?
2. Section 4.3 is particularly interesting to me. However, I think the attacker can easily break the controllability only by changing the order of the text instruction. Do you have any potential solutions and ideas on that?
3. Instead of simple image classification, are there other qualitative examples of counterfactual controls (e.g., VQA)?
4. Any analysis on which set of layers L to intervene on for visual / cross-modality / multimodal editor?
5. What are some insights on fine-tuning only prefix/suffix tokens of textual embedding in the multimodal editor?
6. Minor Errata
   1. L84: hu2024bliva looks typo.

---

> ### Author Response · Authors · 2024-11-19
> **To Reviewer Yzen (Part I)**
>
> Dear Reviewer Yzen,
>
> We sincerely appreciate the time and effort you've devoted to reviewing our work and providing helpful feedback!
>
> **Q1: Regarding the plan to automate the rank-tuning process.**
>
> **A1:** Thank you for the great insights. We completely agree that manual rank searching can be highly time-consuming. To address this, we plan to explore automated rank selection methods in the future. Specifically, we aim to utilize meta-learning frameworks [ref1], which leverage prior training experiences across tasks to efficiently adapt ranks, and Bayesian-based frameworks [ref2], which use probabilistic models to iteratively explore and dynamically select optimal ranks. These approaches will help mitigate the labor-intensive manual searches.
>
> **Q2: Regarding MRT’s contribution.**
>
> **A2:** Thank you for the question. We would like to highlight several key contributions of MRT besides extending the effectiveness of representation tuning into MLLMs.
>
> **First, intuitive yet effective control.** MRT is the first attempt to enable token-wise control over LMMs through representation editing. By directly editing the semantic information of the image RoI and the textual target class indicator token, MRT offers an interpretable and intuitive mechanism for adjusting model predictions. This level of fine-grained controllability is difficult to achieve with existing baselines.
>
> **Second, representation learning loss optimization.** From an optimization perspective, we provide a detailed analysis of why MRT outperforms other PEFT methods. By visualizing the loss landscape, we demonstrate that multimodal representation tuning enhances the generalization capabilities of LMMs, highlighting a promising direction for future PEFT research.
>
> **Third, joint multimodal learning.** Unlike single-modality research, multimodal settings require consideration of two additional factors: **multimodal integration** and **vision modality editing**. To address this, we designed a framework that optimizes the cross-modality layer to effectively bridge the gap between the two modalities. While current PEFT approaches [ref13] for LMMs typically unfreeze the cross-modality projector during stage-2 tuning, we adhere to the principle of representation editing by introducing a lightweight cross-modality editor, achieving significantly lower parameter usage while delivering substantial performance gains. For vision modality editing, MRT takes a markedly different approach from current NLP practices by focusing on editing all visual representations. This method highlights the sparsity of visual information and suggests that broader editing strategies should be explored in the vision domain.
>
> Thank you again for the great suggestion. We have supplemented the above discussions in Appendix S11.
>
>
> **Q3: Regarding mitigation strategies of potential misuse of MRT.**
>
> **A3:** Thank you for the excellent question. We want to answer this question from two perspectives.
>
> **Possible misuse of MRT utilizing controllability.**
> Misuse of generative models can lead to significant ethical, social, and security concerns, especially when the model’s controllability is leveraged maliciously via tuning. There are possible misuse scenarios and corresponding mitigation strategies. First, attackers can manipulate models to produce misinformation (e.g., misclassification) via intentionally altering the model’s understanding of an input image [ref3]. Second, biased information can be produced or amplified. Attackers can edit the textual tokens related to sensitive attributes in the multimodal representation, leading to harmful or discriminatory outputs [ref4].
>
> **How to mitigate potential misuse?**
> In order to mitigate possible misinformation, we suggest performing adversarial robustness testings [ref5] that explicitly check for consistency in object recognition across varying queries. For mitigating bias generation or amplification, one solution can be bias detection and correction procedure on generated content, monitoring the representation for bias patterns and applying corrective measures if detected [ref4]. Another solution lies in clearly documenting any controlled editing made to the model’s representation and disclosing any potential biases introduced during this process [ref6]. In conclusion, while MRT exhibits strong output controllability, applying MRT to realistic applications still requires ethical safeguarding, robust testing, and transparency measuring. From a security perspective, MRT presents significant potential, as it may facilitate the development of white-box attack and defense strategies tailored to LMMs [ref7].
>
> We have added additional discussions to the revised paper (see Appendix S10).

---

> ### Author Response · Authors · 2024-11-19
> **To Reviewer Yzen (Part II)**
>
> **Q4: Changing the order of text instruction can break the controllability.**
>
> **A4:** We would like to clarify that simply changing the order of text instruction can’t break the controllability. MRT is able to accommodate variations in prompt formats by training control editors specifically for the new prompt structure. To validate this, we conducted another controllability experiment where we changed the textual prompt format from “Is the object an e in the image?” to “Is the object in the image an e?”, and trained control editors under the same settings described in Sec. 4.3. The results, as shown in the table below, demonstrate that the new editors achieve equally effective and robust control over counterfactual outputs.
>
> | Class e | Misclassification rate on e | Misalignment rate on e |
> | ---- | ----------- | --------------- |
> | dog  |    100%        |     100%       |
> | cat |    100%      |     100%       |
> | Ship |   100%     |     100%       |
> | Frog |    100%    |    100%        |
> | Ship |   100%     |    100%        |
>
> To further enhance the robustness of MRT’s controllability, one possible solution is to leverage the power of prompt engineering [ref14], which crafts effective prompts to guide the output to generalize the control across an even broader range of input queries, reducing possible sensitivity to variations in phrasing. This involves normalizing different prompts with the same semantic meaning into standardized templates, such as converting “Could you help me identify the object’s name?” or “What’s the name of the object in the image?” into a common structure like “What is the object’s name?”. By standardizing the input prompts, MRT is able to achieve more robust token-wise control, even in scenarios with diverse or complex textual instructions.
>
>
>
>
>
> **Q5: Regarding the qualitative examples of counterfactual controls.**
>
> **A5:** We have included more qualitative examples and experimental results on the Text-VQA dataset in the revised paper. Please refer to Appendix S6.2, thank you!
>
>
> **Q6: Analysis on the set of layers to intervene.**
>
> **A6:** Thank you for your suggestion! Table 3 in Sec. 4.4 illustrates the impact of different editing depths. We would like to further discuss the results here. First, the results indicate MRT’s performance is positively correlated with editing depth. Second, we observe that even under the setting of editing the first layers of the vision encoder and the LLM (i.e., VPT shallow in prompt tuning), MRT still can be able to gain a noticeable improvement (i.e., 1329.84 on MME and 60.57 on MMAvg) in performance. Third, editing only the latter half of the layers yields better performance compared to editing the first half (i.e., 1447.41 vs. 1440.32 on MME), suggesting that deeper layers play a more significant role in enhancing the model's capabilities. Last, editing at every odd layer outperforms both the "first half" and "latter half" configurations (i.e., 1468.21 vs. 1447.41 on MME). This suggests that distributing interventions across the model in a sparse manner can be more beneficial than focusing on a continuous block of layers.
>
> **Q7: Insights on fine-tuning only prefix/suffix tokens.**
>
> **A7:**  It is a great question! During instruction tuning, we fine-tune the prefix and suffix tokens from the textual-oriented representations. Considering the attention mechanism and the generation process of Transformer-based decoder models, prefix segments often condition the model on specific tasks or behaviors, therefore they are crucial for setting up the context and guiding the generation process early on [ref8, ref15]. Suffix segments also play an important role in guiding and controlling generations due to the autoregressive mechanism. Recognizing their importance during training, we focus on these tokens’ editing from the textual embedding. As for the vision encoder, we fine-tune the whole visual sequence since the model relies on the entire visual sequence to capture global context. We have included additional experiments on editing different segments of textual-oriented representations. The results clearly demonstrate that fine-tuning both the prefix and suffix tokens yields the best performance, significantly outperforming the setting of fine-tuning all tokens. Specifically, we observe a substantial drop in the MME score when the entire textual embedding is edited (i.e., 1233.90 vs. 1580.40 on MME). This suggests that over-editing the embeddings can lead to response drift, negatively impacting performance. This observation aligns with recent studies on prompt tuning [ref9-13], which indicate that larger adjustments (i.e., longer inserted prompts in prompt tuning) do not necessarily lead to better performance and can, in fact, be less effective than smaller edits.
>
> | Segments  | MME |
> | ---- | ----------- |
> | Prefix Only  |   1465.32     |
> | Suffix Only |    1497.35    |
> | Prefix & Suffix    |  1580.40      |
> | All         |  1233.90      |

---

> ### Author Response · Authors · 2024-11-19
> **To Reviewer Yzen (Part III)**
>
> **Q8: Typos.**
>
> **A8:** Thank you for pointing it out. We have fixed it accordingly in the revised version.
>
>
> [ref1] Zhang, R., et al. AutoLoRA: Automatically Tuning Matrix Ranks in Low-Rank Adaptation Based on Meta Learning. ArXiv, 2024.
>
> [ref2] Moe, C., et al. Bayesian-LoRA: LoRA based Parameter Efficient Fine-Tuning using Optimal Quantization levels and Rank Values trough Differentiable Bayesian Gates. ArXiv, 2024.
>
> [ref3] Chen, Xiangning, et al. Robust and accurate object detection via adversarial learning. CVPR, 2021.
>
> [ref4] D'Incà, Moreno, et al. OpenBias: Open-set Bias Detection in Text-to-Image Generative Models. CVPR, 2024.
>
> [ref5] Dong, Ziyi, et al. Adversarially-aware robust object detector. ECCV, 2022.
>
> [ref6] Shah, Milind, et al. A Comprehensive Review of Bias in Deep Learning Models: Methods, Impacts, and Future Directions. Archives of Computational Methods in Engineering. ArXiv, 2024.
>
> [ref7] Daizong Liu, et al. Attacks on Large Vision-Language Models: Resources, Advances, and Future Trends. ArXiv, 2024.
>
> [ref8] Bavarian, Mohammad, et al. Efficient training of language models to fill in the middle. ArXiv, 2022.
>
> [ref9] Lester, Brian, et al. The power of scale for parameter-efficient prompt tuning. EMNLP, 2021.
>
> [ref10] Cheng Han, et al. Facing the Elephant in the Room: Visual Prompt Tuning or Full Finetuning? ICLR, 2024.
>
> [ref11] Changdae Oh, et al. Blackvip: Black-box visual prompting for robust transfer learning. ArXiv, 2023.
>
> [ref12] Chengzhi Mao, et al. Doubly right object recognition: A why prompt for visual rationales. ArXiv, 2022.
>
> [ref13] Taowen Wang, et al. M$^2$PT: Multimodal prompt tuning for zero-shot instruction learning. EMNLP, 2024.
>
> [ref14] White, J., et al. A prompt pattern catalog to enhance prompt engineering with chatgpt. ArXiv, 2023.
>
> [ref15] Raffel, Colin, et al. Exploring the limits of transfer learning with a unified text-to-text transformer. Journal of machine learning research 21.140 (2020): 1-67.
>
> We sincerely appreciate your thoughtful comments. We hope our response addresses your concerns. Please let us know if there are any additional questions, and we will be happy to discuss further.

---

> > ### Comment · Reviewer_Yzen · 2024-11-22
> >
> > Thanks for the detailed responses and additional results. Most of my concerns are well addressed. I think the mentioned contribution in A2 is intuitive but not novel enough to increase the score. I hope to see some generalization results leveraging LLM regarding Q4. I will keep the original score.

---

> > > ### Author Response · Authors · 2024-11-23
> > > **Thank you for the prompt response**
> > >
> > > Thank you for your valuable feedback.
> > >
> > > To further address your comment on generalization, we have leveraged **a lightweight rephraser** based on T5-small (i.e., 60M parameters), and customized a dataset for fine-tuning the rephraser, containing **6 different variant templates** with the same semantic meaning of the expected input sequence. The table below shows that MRT can successfully achieve robust control on various input sequences with a single set of editors, even if they differ in lengths and structures.
> > >
> > > | Input Variants | Output Control Rate on e |
> > > | -------------------------------------------------------- | ----------- |
> > > | “Is there an e object visible in the image?”  |    100%        |
> > > | "Does the image contain an object that is an e?" |    100%      |
> > > | "Is the object shown in the image an e?" |   100%     |
> > > | "Is the object in the picture an e?" |    100%    |
> > > | "Do you recognize the object in the image as an e?" |   100%     |
> > > | "Can you tell if the object shown in the image is specifically an e?" |   100%     |
> > >
> > > In addition, we further evaluate the generalization of MRT via testing its controllability on the Text-VQA dataset, which derives different counterfactual controls from the current image classification settings. Specifically, we select 8,017 instances as the training set and 1,189 instances as the validation set on textual tokens beginning with “what is the $n$”, where $n$ represents an image attribute (e.g., name, color, brand). We aim to generate counterfactual outputs different from the scenarios (i.e, misclassification and misalignment) of counterfactual outputs on image classification task, we target the scenario of **Indeterminate** by altering the labels of all questions related to $n$ in the training set to “Not sure”. The results show that MRT can control across an even broader range of input queries.
> > >
> > > | Attribute ($n$) | Indeterminate |
> > > | -------- | ---------------- |
> > > | name  |    100%        |
> > > | color  |    100%        |
> > > | brand  |    100%        |
> > >
> > > Altogether, MRT demonstrates consistency and effectiveness of its control strategy for textual inputs. We have added additional discussions as well as qualitative results to the revised paper Appendix Sec. S6. Thank you, and we are eager to address any of your remaining concerns during the discussion phase.

---

### Official Review · Reviewer_uVrb · 2024-11-05

**Soundness:** 3
**Presentation:** 3
**Contribution:** 3
**Rating:** 6
**Confidence:** 4

**Summary:**

The paper proposes a novel Multimodal Representation Tuning which can editing LMM representation and provide control. The paper introduces a representation editor $\phi$ based on linear representation hypothesis and interchange interventions, which can apply to different representations in LMM. The overall writing is clear. The experiments cover the comparison between the MRT and the other PEFT methods.

**Strengths:**

1.	The overall writing is clear.
2.	The idea is general and could be applied to many different applications. I think this would be of interest to people in the LMM community.
3.	The proposed method is simple yet effective.
4.	The experiments clearly show the improvement of the MRT over the PEFT.

**Weaknesses:**

1.	There are some typos. For example, ##hu2024bliva## in the "Multimodal Instruction Tuning" section of the related work.
2.	The ablation study is insufficient. I would expect more ablation experiments, such as applying MRT only to Visual Representation and Cross-modality Representation.
3.	There is too little theoretical analysis on why MRT is better than PEFT.

**Questions:**

Please see Weaknesses.

---

> ### Author Response · Authors · 2024-11-19
> **To Reviewer uVrb**
>
> Dear Reviewer uVrb,
>
> We sincerely appreciate your time and effort in reviewing our paper and providing valuable comments. We provide explanations to your questions point-by-point in the following.
>
> **Q1: Regarding the typos.**
>
> **A1:** Thank you for pointing it out. We have revised accordingly.
>
> **Q2: Regarding more ablation experiments.**
>
> **A2:** Following the suggestion, we have conducted additional experiments of applying MRT to individual components one at a time. The results are summarized in the table below. Notably, the best performance is achieved when representation tuning is applied simultaneously to all components of the base LMM model. This aligns with our ablation study (Figure 6, left), which demonstrates that removing any single component from MRT results in a noticeable performance decline (e.g., a score of 1376 on MME when the visual editor is excluded). These results, along with additional discussions, have been included in the revised paper (see Appendix S4).
>
> | Applied Component  | MME | MMAvg |
> | ---- | ----------- | --------------- |
> | LLM  | 1473.25        | 62.90            |
> | Cross-modality | 1165.33        | 53.67            |
> | Vision encoder | 1342.46        | 60.83            |
> | All | 1580.40        |   64.93     |
>
> **Q3: Regarding more theoretical analysis on why MRT is better than other PEFT.**
>
> **A3:** Thank you for the great suggestion. In our work, we have conducted the optimization analysis based on loss landscape [ref1-2], indicating that MRT gains a flatter loss landscape, which in turn provides more optimization choices compared to other PEFT methods [ref3-4]. Additionally, our token-wise control experiment on visual RoI and textual target indicator shows that precise control over specific token representations can effectively alter the model behavior. Following your suggestion, we plan to conduct further theoretical analysis, including how the incorporation of representation editing influences the attention module (e.g., attention activation pattern analysis [ref4]) and gradient flow analysis [ref5]. We have also highlighted this as a future direction in our revised paper (see Appendix S11).
>
> [ref1] Hao Li, et al. Visualizing the loss landscape of neural nets. NeurIPS, 2018.
>
> [ref2] Runjia Zeng, et al. Visual Fourier Prompt Tuning. NeurIPS, 2024.
>
> [ref3] Ying Shen, et al. Multimodal Instruction Tuning with Conditional Mixture of LoRA. ACL, 2024.
>
> [ref4] Taowen Wang, et al. M$^2$PT: Multimodal prompt tuning for zero-shot instruction learning. EMNLP, 2024.
>
> [ref5] Bambhaniya, et al. Progressive Gradient Flow for Robust N: M Sparsity Training in Transformers. 2024.
>
> We sincerely appreciate your thoughtful comments. We hope our response addresses your questions. Please let us know if there are any additional questions, and we will be happy to discuss further.

---

> > ### Author Response · Authors · 2024-11-29
> > **Looking forward to the discussion**
> >
> > Dear Reviewer uVrb,
> >
> > We sincerely appreciate your dedicated time and effort in reviewing our submission. We understand how demanding your schedule might be and are genuinely grateful for your valuable insights. As the discussion phase nears its conclusion, we kindly seek your input on our responses to your feedback.
> >
> > We hope to confirm that we’ve adequately addressed your concerns and are open to discussing any remaining points or questions you may have. Your input is invaluable to us, and we would greatly value the chance to discuss them with you.
> >
> > Thank you again for your time and consideration.
> >
> > Best regards, \
> > The Authors

---

> > > ### Comment · Reviewer_uVrb · 2024-12-03
> > >
> > > I appreciate the authors' feedback and the new experimental results, which address part of my concerns. Overall, based on the revised version, the experimental results are now quite comprehensive. The method is relatively simple, but the results demonstrate its effectiveness. However, I still believe that the insights and contributions brought by MRT, compared to previous methods like PEFT, are somewhat limited. From my perspective, I would prefer to see the multimodal learning community focus on developing more novel approaches to tackle more complex problems. Therefore, I will maintain my original score.

---

> > > > ### Author Response · Authors · 2024-12-03
> > > > **Thank you for the response**
> > > >
> > > > We sincerely appreciate your time and effort in reviewing our paper and providing valuable feedback, which is essential for improving the work. We are glad that our responses have addressed most of your concerns and that you've maintained the positive assessment of our work.

---

### Author Response · Authors · 2024-11-20
**Summary of Revisions**

To all reviewers:

Thank you for your thorough review and insightful comments. We have revised our paper according to the suggestions. The major changes are summarized as follows:

* We have performed more ablation experiments to explore applying MRT to a single modality at a time, detailed in Appendix Sec. S4. We have also further discussed the future plan to conduct more theoretical analysis of MRT in Appendix Sec. S11. Additionally, we corrected the typo on Line 84 as suggested by **Reviewer uVrb**.

* As suggested by **Reviewer Yzen**, we have added a discussion on prefix/suffix editing in Appendix Sec. S2, supplemented analysis of editing depth in Sec. 4.4, and provided more qualitative examples of counterfactual controls on the Text-VQA dataset in Appendix Sec. S6. Additionally, we discussed MRT's robustness and generalizability in Appendix Sec. S6, addressed potential misuse and mitigation in Appendix Sec. S10, and corrected the typo on Line 84. Key technical contributions have also been highlighted in Appendix S11.

* Based on the suggestions by **Reviewer PDvR**, we have included more discussion on the robustness of MRT’s controllability in Appendix Sec. S6 and reduced redundancy in Sec. 3.2. In addition, we have added more discussion on the optimization landscape in Appendix S11.

* We conducted additional experiments evaluating MRT on SEED and GQA datasets, as well as applying MRT to LMMs with different vision encoders and LLMs, presented in Appendix Sec. S3. Training time and memory efficiency comparisons were included in Appendix Sec. S5, and the visual representation selection procedure was clarified in Sec. 3.2. We have performed more controllability experiments in Appendix Sec. S6. Moreover, we have also highlighted our key technical contributions in Appendix S11, as suggested by **Reviewer RZ9b**.

All modifications have been marked in ${\color{blue} blue}$ in our revised submission.


Sincerely yours,\
Authors.

---

### Author Response · Authors · 2024-11-22
**Looking forward to the discussion**

Dear Reviewers,

We sincerely appreciate the time and effort you've devoted to reviewing our work. We understand that your schedule may be quite busy, and we are truly grateful for your valuable feedback. As we are presently in the discussion phase, we would greatly value the opportunity to engage in further dialogue with you. Our aim is to gain insights into whether our responses effectively address your concerns and to ascertain if there are any additional questions or points you would like to discuss.

We look forward to the opportunity for further discussion with you. Thank you for your thoughtful consideration.

Best regards,

Authors

---

### Author Response · Authors · 2024-12-02
**Summary of Author-Reviewer Discussion**

Dear Area Chair and Reviewers,

We would like to express our sincere gratitude for your efforts in facilitating the discussion regarding our paper. As the discussion is coming to an end, we would like to provide a brief summary of the key points that have been discussed:

- We have included ablation studies to explore applying MRT to a single modality at a time and discussion of the future plan to conduct more theoretical analysis of MRT, as suggested by **Reviewer uVrb**.

- In response to **Reviewer Yzen**'s suggestions, we have included a discussion on the prefix/suffix editing of textual-oriented representations and analysis of editing depth. We have also conducted thorough experiments on the extension to VQA task, robustness, and generalizability (i.e., prompt engineering) of MRT’s controllability. Moreover, we have discussed the potential misuse and mitigation of controlling model output and highlighted three key technical contributions of MRT.

- We have provided more discussion on the distinction of MRT’s counterfactual output control compared to other PEFT methods and robustness of MRT’s controllability with different prompt templates, based on **Reviewer PDvR**’s suggestions. Additionally, we have reduced the redundancy of the main method section and included more discussion on the optimization landscape.

- We have conducted additional experiments to evaluate MRT on SEED and GQA datasets, alongside applying MRT to LMMs with different vision encoders and LLMs and evaluating the generalizability of our method under different training datasets. We have also included more experimental results on the extension, robustness and generalizability of output control, and highlighted our key contributions. In addition, we have clarified the visual representation selection procedure and the choice of tuning prefix/suffix tokens on textual-oriented representations, as suggested by **Reviewer RZ9b**.

In summary, we would like to express our appreciation to all reviewers for acknowledging our responses. We are particularly grateful that Reviewer RZ9b has increased their score, and Reviewer uVrb, Yzen and PDvR have maintained their positive assessment. Although we understand that Reviewer RZ9b has not engaged in subsequent discussion due to the busy schedule, we believe that our response has effectively addressed the concerns through clear explanations and additional experimental results.

We would like to emphasize the contributions of our work, which have been acknowledged by the reviewers and are important to the community:

***Efficiency and Effectiveness:*** Existing PEFT approaches for LMMs tend to focus on reparameterization or adding additional heavy layers or trainable prompts, overlooking the potential of semantically rich multimodal representations. In contrast, MRT pioneers a fundamentally different approach by concentrating on directly editing these representations. Our approach eliminates the need for extensive parameter usage or complicated architectural modifications, making it significantly more computationally efficient. Despite utilizing only 0.03% of the tunable parameters, MRT achieves state-of-the-art performance across various benchmarks, proving that manipulation of multimodal representations can deliver both efficiency and effectiveness. This balance of reduced computational demands with high performance underscores MRT's impact on PEFT strategies for LMMs.

***Multimodal Integration:*** Our approach utilizes editing on semantically rich multimodal representation, considering not only multi-modality editing but multimodal integration. Unlike existing PEFT approaches on LMMs which either keep the cross-modality layer frozen or make it fully trainable, we introduce a lightweight cross-modality representation editor. This innovative design efficiently bridges the gap between visual and textual modalities, enabling effective alignment while maintaining computational efficiency.

***Controllability and Interpretability:*** MRT is the first attempt to enable token-wise control over LMMs through representation editing. By directly editing the semantic information of the image RoI and the textual target class indicator token, MRT offers an interpretable and intuitive mechanism for adjusting model predictions. This level of fine-grained controllability is difficult to achieve with existing baselines. Additionally, we provide a detailed analysis of why MRT outperforms other PEFT methods via visualizing the loss landscape, demonstrating that multimodal representation tuning enhances the generalization capabilities of LMMs.

Finally, we deeply value the constructive comments provided by the reviewers. In response, we have carefully refined our work based on the feedback received. Considering the contributions made, we hope that our work can provide new insights to the efficient tuning communities, and contribute to their further development.

Sincerely,\
Authors

---

### Meta-Review · Area_Chair_nuB1 · 2024-12-21

**Metareview:**

The paper introduces Multimodal Representation Tuning (MRT), a new parameter-efficient fine-tuning (PEFT) method for large multimodal models (LMMs). Instead of inserting additional trainable parameters into each layer, MRT directly edits and refines the latent representations at various points in the vision, language, and cross-modal processing stacks. The authors argue that focusing on representation-level adjustments offers several advantages: improved interpretability, fine-grained controllability over model outputs, and better performance than existing PEFT techniques with fewer tunable parameters.

Strengths:

-- MRT uses significantly fewer parameters than competing approaches yet maintains or exceeds performance. This resource-friendliness is appealing for real-world scenarios with limited computational budgets.

-- By editing representations directly, MRT enables counterfactual interventions and more transparent ways to understand how multimodal information flows and influences the final output.

-- The authors present experiments on diverse tasks and show consistent gains over existing baselines. The clear improvement across multiple multimodal benchmarks supports the robustness of the approach.

Weaknesses:

-- While the paper includes some ablation studies, reviewers note that further investigations—e.g., isolating edits on visual vs. cross-modal representations—could strengthen the findings. Additionally, performance on larger, more commonly used datasets (e.g., VQAv2, DocVQA) would bolster claims of general applicability.

-- The paper mainly provides empirical evidence but offers limited theoretical rationale for why representation editing yields these gains. Moreover, the important rank parameter is hand-tuned, leaving open the question of how to automate that process for wider adoption.

-- MRT’s controllability demonstrations are intriguing but somewhat narrow. More varied examples—especially across different NLP tasks or more complex queries—would strengthen the case for how effectively MRT can be leveraged in practice without risking misuse or adversarial manipulation.

Overall, the paper makes a notable contribution to parameter-efficient fine-tuning for LMMs. MRT’s approach to editing multimodal representations is clearly distinctive and yields strong empirical improvements. The reviewers appreciate the method’s simplicity, interpretability, and the strong reported results.  The AC agrees with the majority of the reviewers to accept the paper.

**Additional Comments On Reviewer Discussion:**

The weaknesses are described above. The authors have addressed most comments in rebuttal and most of the reviewers generally agree to accept the paper.

---

### Decision · Program_Chairs · 2025-01-22

Accept (Poster)